# Traceable Evidence Enhanced Visual Grounded Reasoning: Evaluation and Method

**Haochen Wang**[1,2]  **Xiangtai Li**[3]  **Zilong Huang**[3]  **Anran Wang**[3]  **Jiacong Wang**[2,3]  **Tao Zhang**[3]
**Jiani Zheng**[3]  **Sule Bai**[3]  **Zijian Kang**[3]  **Jiashi Feng**[3]  **Zhuochen Wang**[3]  **Zhaoxiang Zhang**[1,2*]

[1]New Laboratory of Pattern Recognition (NLPR),
 State Key Laboratory of Multimodal Artificial Intelligence Systems (MAIS),
 Institute of Automation, Chinese Academy of Sciences (CASIA)
[2]University of Chinese Academy of Sciences  [3]ByteDance
 `{wanghaochen2022, zhaoxiang.zhang}@ia.ac.cn`

## Abstract

Models like OpenAI-o3 pioneer visual grounded reasoning by dynamically referencing visual regions, just like human "thinking with images". However, no benchmark exists to evaluate these capabilities holistically. To bridge this gap, we propose **TreeBench** (Traceable Evidence Evaluation Benchmark), a diagnostic benchmark built on three principles: (1) *focused visual perception* of subtle targets in complex scenes, (2) *traceable evidence* via bounding box evaluation, and (3) *second-order reasoning* to test object interactions and spatial hierarchies beyond simple object localization. Prioritizing images with dense objects, we initially sample 1K high-quality images from SA-1B, and incorporate eight LMM experts to manually annotate questions, candidate options, and answers for each image. After three stages of quality control, **TreeBench** consists of 405 challenging visual question-answering pairs, even the most advanced models struggle with this benchmark, where none of them reach 60% accuracy, *e.g.*, OpenAI-o3 scores only 54.87. Furthermore, we introduce **TreeVGR** (Traceable Evidence Enhanced Visual Grounded Reasoning), a training paradigm to supervise localization and reasoning jointly with reinforcement learning, enabling accurate localizations and explainable reasoning pathways. Initialized from Qwen2.5-VL-7B, it improves V* Bench (+16.8), MME-RealWorld (+12.6), and **TreeBench** (+13.4), proving traceability is key to advancing vision-grounded reasoning. The code is available at https://github.com/Haochen-Wang409/TreeVGR.

## 1 Introduction

Recent breakthroughs in Large Language Models (LLMs) reasoning, such as OpenAI-o1 (OpenAI, 2024b) and DeepSeek-R1 (Guo et al., 2025a) with remarkable test-time scaling properties, have motivated researchers to explore reasoning for Large Multimodal Models (LMMs) (Huang et al., 2025; Wei et al., 2025a;b; Chen et al., 2025). These models are typically remarkable in their mathematical and scientific reasoning, particularly through *text-space* reasoning. However, they exhibit critical limitations when applied to perception-heavy tasks (Jiang et al., 2025) or general multimodal benchmarks (Wang et al., 2024c), primarily due to accumulated language bias from their exclusive reliance on textual reasoning pathways. A paradigm shift toward *visual grounded reasoning* emerged with models like OpenAI-o3 (OpenAI, 2025), which is able to "think with images" by dynamically referencing and amplifying task-relevant regions during reasoning, resulting in *image-text interleaved* reasoning pathways. Yet, despite growing interest, the community currently lacks comprehensive evaluation benchmarks for assessing these capabilities.

Classical benchmarks like POPE (Li et al., 2023c), MMBench (Liu et al., 2023b), SEED-Bench (Li et al., 2023a), and MMMU (Yue et al., 2024) usually overlook fine-grained localization and verifiable reasoning chains. Others (Wu & Xie, 2024; Zhang et al., 2024a; Wang et al., 2025f; Dong et al., 2024; Wang et al., 2025b;a; Zhang et al., 2024b) *partially* address localization but lack traceability

---
*Corresponding author.

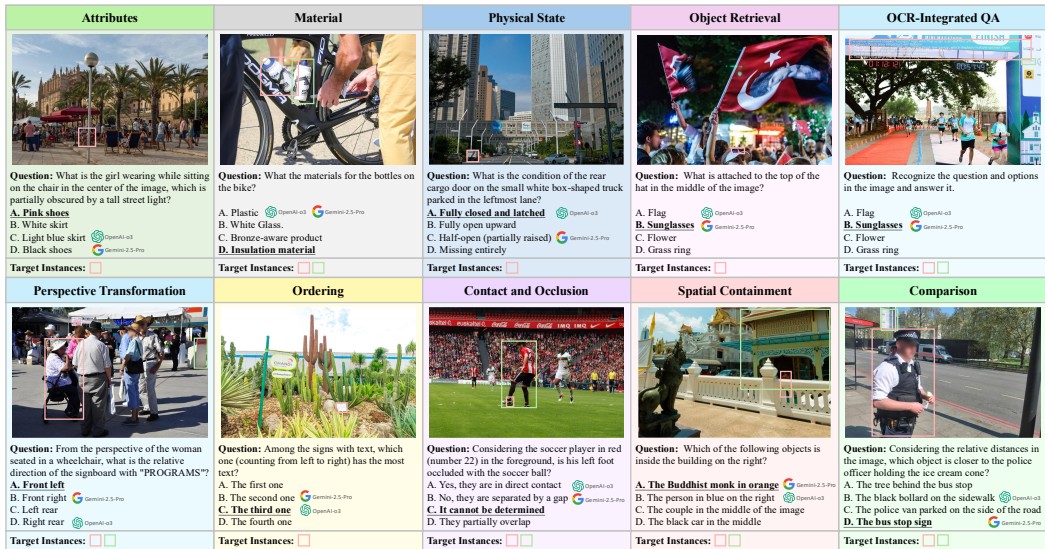

Figure 1: Qualitative examples from **TreeBench** for each discipline. Each question requires focused visual parsing on mere objects, and some even request second-order reasoning beyond precise localization. Moreover, the bounding boxes of all target instances are provided, ensuring a traceable evaluation. All these questions are challenging, as OpenAI-o3 (OpenAI, 2025) and Gemini-2.5-Pro (DeepMind, 2025b) *cannot* answer them correctly simultaneously.

or complex reasoning: V* Bench (Wu & Xie, 2024) is restricted to simple spatial queries (*e.g.*, "Is A left of B?") and risks data contamination with COCO-derived images (Lin et al., 2014); MME-RealWorld (Zhang et al., 2024a), HR-Bench (Wang et al., 2025f), and document benchmarks (Biten et al., 2022; Mathew et al., 2021; Liu et al., 2023c) support high-resolution inputs but lack traceable evidence and second-order reasoning such as perspective shifts. In short, these benchmarks fail to adequately evaluate three key elements central to visual grounded reasoning: nuanced visual grounding, traceable multi-step reasoning, and dynamic cross-modal interaction through *interleaved* box-text reasoning pathways.

To bridge this gap, we propose **TreeBench** (Traceable Evidence Evaluation Benchmark), designed around three foundational principles essential for evaluating true "thinking with images" capabilities:

- **Focused Visual Perception.** It evaluates a model's ability to identify *subtle* targets within cluttered, real-world scenes using detailed, precise, and unique textual descriptions, which requires hierarchical scene understanding and the discrimination of extremely similar distractors.

- **Traceable Evidence.** It not only evaluates the final accuracy but also pioneers quantifiable evaluation of reasoning chains, resulting in an explainable, reliable, and transparent evaluation.

- **Vision-Centric Second-Order Reasoning Capabilities.** It moves beyond simple object localization and primitive "what/where" queries. It focuses on complex physical interactions between objects (such as contact and occlusion), as well as spatial containment (inside/outside, above/below) and relative relationships with perspective transformation.

To conduct **TreeBench**, we sample 1K images from SA-1B (Kirillov et al., 2023), prioritizing images with dense objects, as SA-1B (Kirillov et al., 2023) offers high-resolution, real-world scenes with many small and varied objects, making it particularly suitable for evaluating visual grounded reasoning. Subsequently, 8 experts with solid technical backgrounds are involved in hand-crafted annotation for 10 sub-tasks, as demonstrated in Figure 1. In particular, we present a semi-automated pipeline. Each of OpenAI-o3 (OpenAI, 2025) and Gemini-2.5-Pro (DeepMind, 2025b) is required to create three distinct questions belonging to a specific subtask, accompanied by multiple-choice options and the respective correct answers. Subsequently, experts curated or replaced these to ensure quality and difficulty. We additionally incorporate a cross-verification stage for further quality control. Finally, **TreeBench** incorporates 405 high-quality and extremely challenging VQA pairs with accurate bounding boxes of target instances. A comprehensive comparison between **TreeBench** and other related benchmarks is provided in Table 1. Key advantages are:

Table 1: Comparison between benchmarks related to "thinking with images". **TreeBench** features traceable evidence annotations, as well as high input resolution and challenging questions.

| Benchmark | Resolution | Traceable Evidence Annotation | Mean Area of Target Objects (↓) | Qwen2.5-VL-72B Performance (↓) |
|---|---|---|---|---|
| V* Bench | 2,246×1,583 | ✗ | – | 85.9 |
| HR-Bench-4K | 4,023×3,503 | ✗ | – | 79.3 |
| HR-Bench-8K | 5,727×4,430 | ✗ | – | 76.0 |
| MME-RealWorld | 2,076×1,434 | ✗ | – | 62.9 |
| **TreeBench** | 2,152×1,615 | ✓ | **3.05%** | **42.2** |

- **Annotation Quality.** Unlike benchmarks relying on LMM-generated labels such as MMT-Bench (Ying et al., 2024) and SEED-Bench (Li et al., 2023a), our expert-driven process ensures correctness and extreme difficulty. However, relying on models would inevitably introduce significant noise, compromising the quality of the annotations. On the contrary, our **TreeBench** is manually designed by 8 LMM experts, ensuring the annotation correctness and ensuring the difficulty of all questions.

- **Small Target Objects.** All questions in **TreeBench** focus on extremely small objects in complex real-world scenes, where target instances occupy an average of 3.05% of the image.

- **Traceable Evidence Evaluation.** Our **TreeBench** provides bounding box annotations of each target instance. It not only evaluates the final answer, but also reveals the quality of *intermediate reasoning steps*. Those predicted bounding boxes serve as a window into its process, helping to diagnose the source of errors, *i.e.*, whether the model misunderstood the question or failed to locate the relevant object.

- **Task Difficulty.** While models approach saturation (>90%) on benchmarks like V* Bench (Wu & Xie, 2024), even open-sourced state-of-the-art performers like Qwen2.5-VL-72B (Bai et al., 2025a) achieve only 42.2 on our **TreeBench**, implying a large potential improvement.

Beyond evaluation, we further introduce **TreeVGR** (Traceable Evidence for Visual Grounded Reasoning), a training paradigm enhancing localization-driven visual reasoning. Previous attempts like (Wang et al., 2025e; Zheng et al., 2025b; Cao et al., 2025; Fan et al., 2025; Shao et al., 2024a; Qi et al., 2024; Su et al., 2025; Liu et al., 2025a) solely supervise final answers and neglect intermediate region-of-interest generation processes. It becomes hard to quantify the actual contribution of the "grounding-then-answering" framework. On the contrary, we propose **TreeVGR**, a novel training methodology emphasizing traceable evidence through reinforcement learning (RL), which *explicitly supervises bounding box generation*.

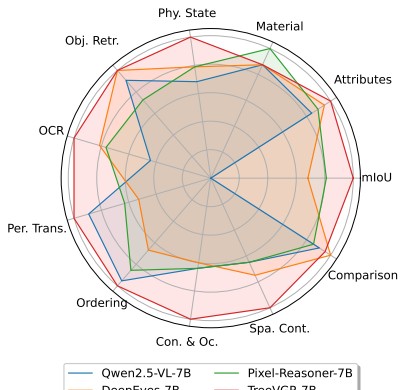

Figure 2: Normalized performance comparison with our **TreeVGR** and other works (Bai et al., 2025a; Zheng et al., 2025b; Su et al., 2025) on our **TreeBench** for each category.

Building on RL with conventional accuracy-based and formatting rewards, **TreeVGR** leverages a novel *dual* IoU reward to ensure both precision and recall in localizing the ground-truth bounding boxes for each target instance. To implement this, we curate 37K samples for RL training, each comprising an image, a question, an answer, and corresponding *bounding box annotations* for all target instances. Empirically, initialized from Qwen2.5-VL-7B (Bai et al., 2025a), **TreeVGR** brings significant improvements on various benchmarks, *i.e.*, +16.8 on V* Bench (Wu & Xie, 2024), +12.6 on MME-RealWorld-Lite (Zhang et al., 2024a), and +13.4 on our **TreeBench**. Moreover, as illustrated in Figure 2, compared with related approaches, our **TreeVGR** enables traceable and explainable reasoning pathways with more accurate localizations (mIoU), and finally contributes to bootstrapped overall performance.

In conclusion, **TreeBench** pioneers the evaluation of how models "think with images", while **TreeVGR** establishes a blueprint for training them. Together, they significantly advance the depth and utility of multimodal reasoning assessment with *traceable evidence*.

## 2 RELATED WORK

**Large Multimodal Models.** Initial breakthroughs in Large Multimodal Models (LMMs), such as Flamingo (Alayrac et al., 2022) and BLIP-2 (Li et al., 2023b), achieved this by integrating visual features into the LLM backbone via cross-attention mechanisms. A significant shift towards efficiency emerged with LLaVA (Liu et al., 2023a), which introduced a much more efficient approach. It projects visual features from a pre-trained encoder (*e.g.*, CLIP (Radford et al., 2021)) directly into the LLM's semantic space using a simple two-layer MLP. This paradigm of feature projection catalyzed rapid advancement. Subsequent research has dramatically scaled LMM capabilities and tackled increasingly complex tasks (Wang et al., 2025c; Liu et al., 2024a; Li et al., 2024; Wang et al., 2025d; Bai et al., 2025a; Wang et al., 2024b; Lei et al., 2025; Zhu et al., 2025; Wu et al., 2024; Wang et al., 2024a; Yang et al., 2024a;b; 2025b;a; 2026). A critical frontier has been handling high-resolution inputs. Models like LLaVA-NeXT (Liu et al., 2024a) and InternVL-1.5 (Chen et al., 2024b) adopt any resolution strategy. Qwen2-VL (Wang et al., 2024b) and Qwen2.5-VL (Bai et al., 2025a) introduce multimodal Rotary Position Embedding (mROPE) to support arbitrary resolution inputs. Beyond resolution, scaling pretraining with high-quality data is also vital, as demonstrated by InternVL3 (Zhu et al., 2025). Collectively, these models represent the state-of-the-art, forming robust baselines for diverse real-world multimodal applications. Our work builds upon these advances by leveraging their strong native visual grounding capabilities. However, existing LMMs do not naturally perform an explicit "grounding-then-answering" process, often resulting in misaligned or incomplete responses. By explicitly modeling this sequential process, our approach ensures more accurate and interpretable answers through grounded reasoning.

**Reasoning LMMs.** The groundbreaking reasoning capabilities of LLMs, exemplified by systems like OpenAI-o1 (OpenAI, 2024b) and DeepSeek-R1 (Guo et al., 2025a) have motivated efforts to extend similar competencies to multimodal settings using reinforcement learning (RL) (Sutton et al., 1998). Early approaches primarily focused on equipping LMMs to solve complex math and science problems involving image inputs (Huang et al., 2025; Wei et al., 2025a;b; Chen et al., 2025). Other approaches (Shen et al., 2025; Liu et al., 2025b; Bai et al., 2025b) directly adopt GRPO (Shao et al., 2024b) to open-ended visual grounding. Moreover, some attempts (Liu et al., 2024b; Mondal et al., 2024; Shao et al., 2024a; Qi et al., 2024) focus on regions-of-interest localization before actually answering the question. A recent milestone, OpenAI-o3 (OpenAI, 2025), advanced multimodal reasoning by enabling dynamic image manipulation, *e.g.*, cropping and zooming into regions of interest, to emulate human-like "thinking with images." Subsequent research has sought to replicate this capability through diverse strategies: constructing SFT data (Wang et al., 2025e), vanilla RL (Fan et al., 2025), framing grounding as a function (Zheng et al., 2025b), decoupling grounding and answering (Cao et al., 2025), multi-task reinforcement learning (Liu et al., 2025a), and curiosity-driven reasoning (Su et al., 2025). Critically, these RL-based methods supervise *only* the final answer. In contrast, our **TreeVGR** emphasizes *traceable evidence* during RL training, *i.e.*, supervising generated bounding boxes to ensure precise localization throughout the reasoning process. By doing so, **TreeVGR** enables more transparent, reliable, and fine-grained control over the reasoning pipeline.

**Benchmarks for LMMs.** Current benchmarks lack comprehensive evaluation of multimodal models' ability to "think with images", a capability demanding three core competencies: (1) focused visual perception (identifying small targets in large scenes), (2) traceable evidence (evaluating generated bounding boxes for explainability), and (3) second-order reasoning (deriving insights *beyond* precise instance localization). Some benchmarks may *partially* satisfy the first condition. While some benchmarks address isolated aspects, critical gaps persist. Classical benchmarks like POPE (Li et al., 2023c), MMBench (Liu et al., 2023b), SEED-Bench (Li et al., 2023a), and MMMU (Yue et al., 2024) usually overlook fine-grained localization and verifiable reasoning chains. V* (Wu & Xie, 2024) evaluates detailed attributes and spatial relationships (*e.g.*, "Is A left of B?") but relies on COCO-derived images (Lin et al., 2014), introducing high contamination risk. MME-RealWorld (Zhang et al., 2024a) and HR-Bench (Wang et al., 2025f) support high-resolution inputs but lack traceable evidence, and their questions often become easy when grounded precisely. Crucially, no benchmark integrates all three requirements, particularly the need for complex reasoning conditional on precise grounding, *e.g.*, perspective transform: "*From the perspective of person A, what is the relative direction of object B?*". To bridge this gap, we propose **TreeBench**, the first benchmark designed explicitly for "thinking with images" with *traceable*, multistep evaluation. Beyond accuracy, **TreeBench** assesses: (1) region quality, *i.e.*, faithfulness of generated regions-of-interest in visual reasoning chains, and (2) second-order reasoning, *i.e.*, capabilities requiring inference *beyond* localization. State-of-the-art

models, Gemini-2.5-Pro (DeepMind, 2025b) and OpenAI-o3 (OpenAI, 2025), perform poorly on **TreeBench** (<60%), underscoring its rigor and the unmet challenges in multimodal reasoning.

## 3 TREEBENCH

**TreeBench** is designed to address a critical gap in multimodal evaluation by establishing the first comprehensive benchmark for assessing "thinking with images" capabilities. Specifically, it mainly evaluates (1) the ability of identifying small target objects with long, detailed, and unique text captions in large, complex, and real-world scenes, (2) the explainability of reasoning pathways and traceable evidence, and (3) second-order reasoning beyond precise localization. Our **TreeBench** systematically evaluates 10 core competencies through 405 distinct questions, organized into two progressive protocols, *i.e.*, "Perception" and "Reasoning", with representative examples in Figure 1. In the following, we provide a detailed exploration of task definitions. The annotation pipeline and the final statistics of **TreeBench** can be found in Appendix B and Appendix C, respectively.

**Perception** evaluates the model's ability to accurately "see" and "identify" specific content, which is one of the basic capabilities of directly extracting and interpreting visual information from every detail of the provided image. These tasks primarily evaluate *first-order* visual reasoning capabilities, where correct answers usually depend on the accurate localization of target questions (*e.g.*, objects, regions, or text) and directly recognize their explicit attributes *without* requiring higher-level logical inference or abstract conceptualization. It includes:

1. **Attributes** evaluates the ability to identify and describe specific visual properties (*e.g.*, color, shape, material, or precise classification) of objects or elements within images, particularly requiring attention to fine details, subtle distinctions, and accurate recognition of small-scale or context-dependent features.

2. **Material** measures the ability to analyze and distinguish material properties (*e.g.*, texture, surface finish, composition, or physical state) through visual cues such as light reflection, transparency, wear patterns, or microscopic structural characteristics, requiring precise reasoning about tactile qualities and material-specific visual indicators.

3. **Physical State** assesses the ability to assess structural integrity (*e.g.*, damage, wear, or breakage), detect positional states (*e.g.*, open/closed, bent/straight), and interpret age-related features (*e.g.*, freshness, decay) through precise analysis of visual cues like cracks, alignment anomalies, lighting/shadow patterns, or contextual degradation markers.

4. **Object Retrieval** probes the ability to interpret linguistically complex, spatially explicit descriptions and map them to visually subtle or contextually embedded targets in images, testing the integration of natural language understanding, spatial grounding, and discriminative object recognition under high specificity constraints.

5. **OCR-Integrated Question-Answering** evaluates the ability to extract text-based questions and answer options from images, requiring seamless integration of OCR, natural language understanding, and multimodal alignment to produce accurate responses grounded in both textual and visual modalities.

**Reasoning** evaluates the ability to analyze and infer meaningful conclusions beyond recognition. These tasks demand *second-order* visual reasoning capabilities, where correct answers require not only accurate localization but also higher-level cognitive operations over aggregated visual evidence. Precise perceptual grounding is just the first step for these tasks. It includes:

1. **Perspective Transform** measures the capacity to perform viewpoint transformations (*e.g.*, aligning viewer-centric and agent-centric frames of reference) and interpret spatial relations under mirror-reversed or perspective-shifted conditions, testing the ability to disambiguate directional relationships that depend on the visualized entity's orientation rather than the image's literal pixel layout.

2. **Ordering** evaluates the ability to analyze linearly ordered arrangements of objects (*e.g.*, left-to-right, front-to-back, or depth-based sequences) and resolve ordinal relationships by integrating spatial context with discriminative feature recognition, requiring precise localization within continuous layouts and contextual comparison of positional cues (*e.g.*, adjacency, centrality, or extremity) to answer questions dependent on sequential alignment and relative placement.

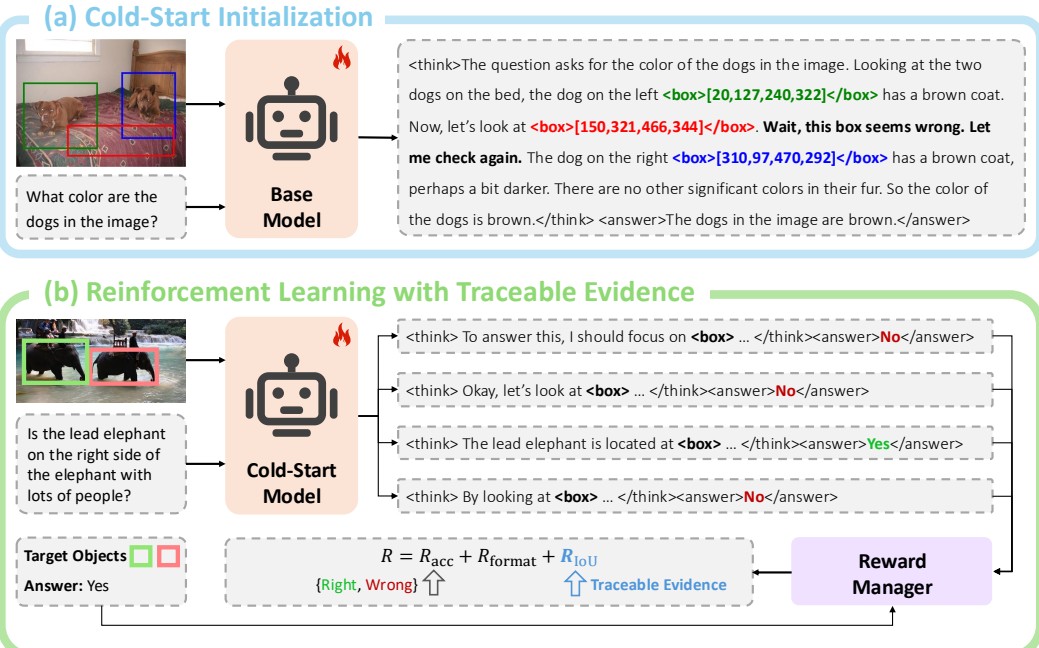

Figure 3: Training pipeline of **TreeVGR**, including (a) a cold-start initialization stage and (b) a reinforcement learning with traceable evidence post-training stage.

3. **Contact and Occlusion** measures the ability to analyze physical interactions between multiple objects (*e.g.*, direct contact, occlusion layers, or shadow-based overlaps) and resolve ambiguities in object identification by leveraging spatial dependencies, requiring precise parsing of contact cues (*e.g.*, alignment, boundary fusion), occlusion boundaries (*e.g.*, partial/full coverage, layer stacking), and contextual constraints to answer questions that hinge on understanding how objects physically coexist and obscure one another in complex scenes.

4. **Spatial Containment** benchmarks the ability to analyze hierarchical spatial relationships (*e.g.*, containment, surface attachment, or regional boundaries) by parsing visual cues like object boundaries, spatial context, and contextual containment rules, requiring precise interpretation of containment hierarchies, surface dependencies, and regional constraints to resolve questions dependent on explicit spatial membership rather than isolated positional attributes.

5. **Comparison** assesses to compare attributes across multiple objects (*e.g.*, distance, size, color) and resolve spatial or perceptual differences, requiring precise parsing of attribute discrimination and contextual distance estimation to answer questions demanding explicit comparison of visually co-present entities.

## 4 TREEVGR

In this section, we introduce our **TreeVGR**. Specifically, we leverage the native grounding capabilities of pre-trained LMMs and unlock *visual grounded reasoning* capabilities, *i.e.*, localizing regions-of-interest first and answering the question next, through a two-stage training pipeline shown in Figure 3, *i.e.*, cold initialization introduced in Section 4.1 and reinforcement learning with traceable evidence elaborated in Section 4.2.

Notably, our **TreeVGR** does *not* require actually replaying cropped images as previous approaches (Wang et al., 2025e; Zheng et al., 2025b; Su et al., 2025) do, as *text-space grounding* is already effective. It leads to much more efficient training and inference procedures.

### 4.1 COLD-START INITIALIZATION

While end-to-end reinforcement learning (RL) has demonstrated validity by (Zheng et al., 2025b) for visual grounded reasoning (VGR) tasks, its practical deployment remains hindered by *extreme*

*computational demands*. Specifically, DeepEyes-7B (Zheng et al., 2025b) requests RL training on 47K samples across *32* episodes, a process requiring 32 H100 (80GB) GPUs operating continuously for 48 hours. Such resource intensity creates barriers to broader accessibility.

To address these limitations, we investigate a computationally efficient alternative. Initial attempts revealed significant training inefficiencies when applying direct RL to VGR: models required extensive iterations to autonomously identify task-relevant visual regions before generating answers. This bottleneck motivates our adoption of a cold initialization strategy as illustrated in Figure 3a. Specifically, we introduce a supervised fine-tuning (SFT) phase using a curated dataset comprising multimodal samples: each sample includes an image, a question, reasoning trajectories with corresponding bounding boxes, and a final answer. This structured initialization ensures VGR capabilities are established prior to RL. Details of data construction and optimization can be found in Appendix E.1.

## 4.2 REINFORCEMENT LEARNING WITH TRACEABLE EVIDENCE

We proceed to reinforcement learning (RL) to refine reasoning trajectories through *traceable evidence supervision* as demonstrated in Figure 3b. Specifically, the bounding boxes generated are evaluated using a box intersection-over-union (IoU) reward, a precise and interpretable metric that measures the alignment between predicted and ground-truth regions. This reward ensures explicit accountability to human-annotated visual evidence, guiding the policy toward spatially accurate and logically coherent reasoning pathways.

**Reward Design.** The total reward consists of three parts: an accuracy reward $R_{\text{acc}} \in \{0,1\}$, a formatting reward $R_{\text{format}} \in \{0,1\}$, and a *dual* Intersection-over-Union (IoU) reward $R_{\text{IoU}} \in [0,1]$:

$$R = R_{\text{acc}} + R_{\text{format}} + R_{\text{IoU}}, \tag{1}$$

where the accuracy reward assesses whether the final answer is correct. We utilize exact-matching for multiple-choice questions, and leverage an online reward model, *i.e.*, Qwen2.5-72B-Instruct (Team, 2024), to judge whether the prediction is correct given the question and the ground-truth answer. The formatting reward ensures the reasoning process and the final answer must be enclosed between `<think>` and `</think>`, and `<answer>` and `</answer>`, respectively. The *dual* IoU reward measures the quality of predicted boxes against ground-truths. Specifically, for $N$ predicted bounding boxes $\{\hat{\boldsymbol{b}}_i\}_{i=1}^N$, where $\hat{\boldsymbol{b}}_i = [\hat{x}_1^i, \hat{y}_1^i, \hat{x}_2^i, \hat{y}_2^i]$ and $M$ ground-truths $\{\boldsymbol{b}_k\}_{k=1}^M$, where $\boldsymbol{b}_k = [x_1^k, y_1^k, x_2^k, y_2^k]$, the *dual* IoU is an average of a *recall* term and a *precision* term.

$$R_{\text{IoU}} = \frac{1}{2}(R_{\text{IoU}}^{\text{R}} + R_{\text{IoU}}^{\text{P}}), \tag{2}$$

where the $R_{\text{IoU}}^{\text{R}}$ indicates the *recall* and $R_{\text{IoU}}^{\text{P}}$ means the *precision*. Specifically, the *recall* term ensures that each ground-truth bounding box $\boldsymbol{b}_k$ is matched with at least one prediction.

$$R_{\text{IoU}}^{\text{R}} = \frac{1}{M} \sum_{k=1}^M \text{IoU}\left[\{\hat{\boldsymbol{b}}_i\}_{i=1}^N, \boldsymbol{b}_k\right], \tag{3}$$

where $\text{IoU}\left[\{\hat{\boldsymbol{b}}_i\}_{i=1}^N, \boldsymbol{b}_k\right] = \max_i \text{IoU}(\hat{\boldsymbol{b}}_i, \boldsymbol{b}_k)$ indicates the maximum IoU between *all* predictions $\{\hat{\boldsymbol{b}}_i\}_{i=1}^N$ and *each* ground-truth $\boldsymbol{b}_k$. Maximizing this term ensures each ground-truth $\boldsymbol{b}_k$ is matched with *at least* one prediction. However, we empirically find that the policy model tends to *enumerate all possible boxes* to obtain a larger recall. Therefore, we introduce a dual term, *i.e.*, $R_{\text{IoU}}^{\text{P}}$, to ensure the *precision* and discourage "empty" boxes that do not match with any ground-truths:

$$R_{\text{IoU}}^{\text{P}} = \frac{1}{N} \sum_{i=1}^N \text{IoU}\left[\{\boldsymbol{b}_k\}_{k=1}^M, \hat{\boldsymbol{b}}_i\right]. \tag{4}$$

Similarly, $\text{IoU}\left[\{\boldsymbol{b}_k\}_{k=1}^M, \hat{\boldsymbol{b}}_i\right] = \max_k \text{IoU}(\boldsymbol{b}_k, \hat{\boldsymbol{b}}_i)$ indicates the maximum IoU between *all* ground-truths $\boldsymbol{b}_k$ and *each* prediction $\{\hat{\boldsymbol{b}}_i\}_{i=1}^N$. Maximizing this term encourages each prediction $\hat{\boldsymbol{b}}_i$ to be matched with *at least* one ground-truth. Therefore, simultaneous optimization of *both* recall and precision eliminates the need for exhaustive enumeration of bounding boxes, thereby contributing to more accurate reasoning pathways. Details of data and optimization can be found in Appendix E.2.

Table 2: Selected results of different models on **TreeBench**. Evaluations of open-source general models are implemented using VLMEvalKit (Duan et al., 2024), while evaluations of visual grounded reasoning models are conducted by us. †Reasoning pathways of o3 (OpenAI, 2025) are unavailable, and thus traceable evaluations are *not* valid. Best performances for open-source models are highlighted in **bold**. *Our* **TreeVGR-7B** *achieves comparable performance with InternVL3-78B* (Zhu et al., 2025).

| | Overall | mIoU | Attributes | Material | Phy. State | Obj. Retr. | OCR | Per. Trans. | Ordering | Con. & Oc. | Spa. Cont. | Comparison |
|---|---|---|---|---|---|---|---|---|---|---|---|---|
| | | | Perception | | | | | Reasoning | | | | |
| **Private Models** | | | | | | | | | | | | |
| Gemini-2.5-Flash-0520 | 45.9 | – | 48.3 | 53.9 | 69.6 | 68.8 | 75.0 | 15.3 | 19.3 | 56.1 | 72.4 | 43.2 |
| GPT-4o-1120 | 46.9 | – | 51.7 | 61.5 | 65.2 | 43.8 | 69.1 | 18.8 | 38.6 | 48.8 | 72.4 | 43.2 |
| Gemini-2.5-Pro-0605 | 54.1 | – | 51.7 | 61.5 | 56.5 | 75.0 | 83.8 | 20.0 | 36.8 | 65.9 | 86.2 | 54.6 |
| o3-0416 | 54.8 | –† | 69.0 | 69.2 | 65.2 | 68.8 | 79.4 | 22.4 | 38.6 | 61.0 | 86.2 | 50.0 |
| **Open-source General Models** | | | | | | | | | | | | |
| LLaVA-OneVision-7B | 37.3 | – | 55.2 | 53.8 | 56.5 | 50.0 | 32.4 | 21.2 | 22.8 | 41.5 | 72.4 | 36.4 |
| LLaVA-OneVision-72B | 40.5 | – | 62.1 | 53.8 | 65.2 | 62.3 | 36.8 | 12.9 | 28.1 | 53.7 | 65.5 | **47.7** |
| Qwen2.5-VL-7B | 37.0 | – | 55.2 | 53.8 | 56.5 | 62.5 | 27.9 | 20.0 | 35.1 | 39.0 | 44.8 | 43.2 |
| Qwen2.5-VL-72B | 42.2 | – | 65.5 | **69.2** | 56.5 | 56.3 | 48.5 | 11.8 | 33.3 | 51.2 | 72.4 | 38.6 |
| InternVL3-8B | 38.8 | – | 51.7 | **69.2** | 56.5 | 56.3 | 33.7 | 21.2 | 24.6 | 39.0 | 72.4 | 43.2 |
| InternVL3-78B | 46.4 | – | 62.1 | 61.5 | 52.2 | **68.8** | 52.9 | 16.5 | 33.3 | 61.0 | **86.2** | 45.5 |
| **Open-source Visual Grounded Reasoning Models** | | | | | | | | | | | | |
| DeepEyes-7B | 37.5 | 30.0 | 62.1 | 53.8 | 65.2 | 68.8 | 51.5 | 11.8 | 24.6 | 36.6 | 51.7 | **47.7** |
| Pixel-Reasoner-7B | 39.0 | 35.7 | 58.6 | 61.5 | 65.2 | 50.0 | 48.5 | 14.1 | 31.6 | 39.0 | 44.8 | 40.9 |
| **TreeVGR-7B** | **50.4** | **44.0** | **65.5** | 53.8 | **82.6** | **68.8** | **63.3** | **22.4** | **36.8** | **61.0** | 69.0 | 45.5 |
| Δ *v.s.* Qwen2.5-VL-7B | ↑13.4 | – | ↑11.7 | −0.0 | ↑26.1 | ↑6.3 | ↑35.4 | ↑2.2 | ↑1.7 | ↑22.0 | ↑24.2 | ↑2.3 |

## 5 EXPERIMENTS

**Baselines.** We include four state-of-the-art private models, GPT-4o-1120 (OpenAI, 2024a) and o3-0416 (OpenAI, 2025) from OpenAI, and Gemini-2.5-Flash-0520 (DeepMind, 2025a) and Gemini-2.5-Pro-0605 (DeepMind, 2025b) from Google. Additionally, representative open-source general models are incorporated, including LLaVA-OneVision series (Li et al., 2024), Qwen2.5-VL series (Bai et al., 2025a), and InternVL3 series (Zhu et al., 2025). Furthermore, two very recent visual grounded reasoning models are also included, *i.e.*, DeepEyes (Zheng et al., 2025b) and Pixel-Reasoner (Su et al., 2025), as both of them follow a "grounding then answering" pipeline, with the capability of "thinking with images". Evaluations are mainly conducted on **TreeBench**, V* Bench (Wu & Xie, 2024), HR-Bench (Wang et al., 2025f), and MME-RealWorld-Lite (Zhang et al., 2024a).

**Results on TreeBench.** Table 2 presents per per-category performance of different models. Overall, OpenAI's o3-0416 (OpenAI, 2025), the state-of-the-art visual grounded reasoning model, demonstrates the strongest perception abilities, as expected. Larger models usually contribute to better performance. Notably, our **TreeVGR-7B** even achieves comparable performance with InternVL3-78B (Zhu et al., 2025), demonstrating the effectiveness of the visual grounded reasoning pipeline. Moreover, compared with visual grounded reasoning models, our **TreeVGR** not only achieves a higher overall performance, but also obtains a larger mIoU, indicating its effectiveness in precisely localizing target objects. More in-depth analysis on **TreeBench** can be found in Appendix D.

**Results on High-Resolution Benchmarks.** In Table 3, **TreeVGR** achieves open-source state-of-the-art on V* Bench (Wu & Xie, 2024). On HR-Bench (Wang et al., 2025f) and MME-RealWorld-Lite (Zhang et al., 2024a) illustrated in Table 3 and Table 4, respectively, our **TreeVGR** brings significant improvements over our base model, Qwen2.5-VL-7B (Bai et al., 2025a). Results on other general benchmarks can be found in Appendix F.1.

**Ablation Studies.** The core contribution of **TreeVGR** is the *traceable* training pipeline, where $R_{IoU}$ is incorporated in conventional RL training. The effectiveness of this design is ablated in Appendix F.2.

Table 3: Comparison with state-of-the-art alternatives on V* Bench (Wu & Xie, 2024) and HRBench (Wang et al., 2025f). All results are self-collected. Best performances of visual grounded reasoning models are highlighted in **bold**.

| | V* Bench | | | HR-Bench-4K | | | HR-Bench-8K | | |
|---|---|---|---|---|---|---|---|---|---|
| | **Overall** | Attr. | Spatial | **Overall** | Single | Cross | **Overall** | Single | Cross |
| **Private Models** | | | | | | | | | |
| GPT-4o-1120 | 66.0 | – | – | – | – | – | – | – | – |
| o3-0416 | 95.7 | – | – | – | – | – | – | – | – |
| **Open-source General Models** | | | | | | | | | |
| LLaVA-OneVision-7B | 70.7 | 73.0 | 60.5 | 64.3 | 74.8 | 53.8 | 59.8 | 65.3 | 54.3 |
| LLaVA-OneVision-72B | 73.8 | 80.9 | 63.2 | 66.3 | 76.5 | 56.0 | 60.9 | 68.8 | 53.0 |
| InternVL3-8B | 72.3 | 73.0 | 71.1 | 70.8 | 79.3 | 62.3 | 62.0 | 64.3 | 59.8 |
| InternVL3-78B | 76.4 | 75.7 | 77.6 | 75.5 | 84.5 | 66.5 | 67.3 | 71.8 | 62.8 |
| Qwen2.5-VL-7B | 74.3 | 77.4 | 69.7 | 72.1 | 88.8 | 55.5 | 68.8 | 83.5 | 54.0 |
| Qwen2.5-VL-72B | 84.8 | 90.8 | 80.9 | 79.4 | 88.8 | 70.0 | 76.3 | 84.3 | 68.3 |
| **Open-source Visual Grounded Reasoning Models** | | | | | | | | | |
| Pixel-Reasoner-7B | 80.6 | 83.5 | 76.3 | 72.9 | 86.0 | 60.3 | 66.9 | 80.0 | 54.3 |
| DeepEyes-7B | 90.0 | 92.1 | 86.8 | 75.1 | **91.3** | 59.0 | 72.6 | **86.8** | 58.5 |
| **TreeVGR-7B** | **91.1** | **94.0** | **87.0** | **77.1** | 90.3 | **64.0** | **73.1** | 86.5 | **59.8** |
| $\Delta$ *v.s.* Qwen2.5-VL-7B | ↑ 16.8 | ↑ 16.6 | ↑ 17.3 | ↑ 5.0 | ↑ 1.5 | ↑ 8.5 | ↑ 4.3 | ↑ 3.0 | ↑ 5.8 |

Table 4: Comparison with state-of-the-art alternatives on MME-RealWorld-Lite (Zhang et al., 2024a). All results are self-collected. The best performance is highlighted in **bold**.

| | | Perception | | | | | Reasoning | | | |
|---|---|---|---|---|---|---|---|---|---|---|
| | **Overall** | OCR | RS | DT | MO | AD | OCR | DT | MO | AD |
| **General Models** | | | | | | | | | | |
| Qwen2.5-VL-7B | 42.3 | 87.6 | 32.7 | 83.0 | 27.3 | 30.0 | 72.0 | 62.0 | 28.7 | 23.0 |
| Qwen2.5-VL-72B | 43.7 | **90.8** | 34.0 | 87.0 | 27.9 | 30.6 | 74.0 | 61.0 | 26.7 | 25.5 |
| LLaVA-OneVision-7B | 43.7 | 80.0 | 40.0 | 56.0 | 31.7 | 39.4 | 65.0 | 33.0 | 38.0 | 32.0 |
| LLaVA-OneVision-72B | 48.7 | 79.2 | 50.7 | 67.0 | 37.9 | 40.0 | 76.0 | 41.0 | 38.7 | 39.3 |
| InternVL3-8B | 47.9 | 83.6 | 49.3 | 75.0 | 34.5 | 36.9 | 70.0 | 44.0 | 40.0 | 37.0 |
| InternVL3-78B | 52.3 | 87.6 | **54.7** | 77.0 | 42.6 | 36.6 | 76.0 | 56.0 | 46.0 | **40.3** |
| **Visual Grounded Reasoning Models** | | | | | | | | | | |
| Pixel-Reasoner-7B | 49.7 | 89.6 | 52.0 | 86.0 | 38.9 | 30.9 | 71.0 | **72.0** | 46.0 | 32.5 |
| DeepEyes-7B | 53.2 | 90.0 | 52.7 | **89.0** | 43.3 | 33.4 | 76.0 | 69.0 | 44.0 | 35.0 |
| **TreeVGR-7B** | **54.9** | 87.6 | 50.7 | 83.0 | **47.0** | **43.4** | 74.0 | 66.0 | **51.3** | 39.0 |
| $\Delta$ *v.s.* Qwen2.5-VL-7B | ↑ 12.6 | – 0.0 | ↑ 18.0 | – 0.0 | ↑ 19.7 | ↑ 13.4 | ↑ 2.0 | ↑ 4.0 | ↑ 22.6 | ↑ 16.0 |

## 6 CONCLUSION

This paper introduces **TreeBench**, a benchmark designed to rigorously evaluate visual grounded reasoning (VGR) or "thinking with images" in large multimodal models, and **TreeVGR**, a two-stage training framework that enhances VGR methods through traceable evidence supervision.

**TreeBench** addresses critical gaps in existing benchmarks by focusing on three principles: focused visual perception (identifying subtle targets in cluttered scenes), traceable evidence (quantifiable reasoning chains via bounding box annotations), and vision-centric second-order reasoning. Constructed through expert-driven annotation and multi-stage quality control, **TreeBench** features 405 high-difficulty visual question-answer pairs with precise bounding boxes, emphasizing small objects in real-world scenarios. It reveals the limitations of state-of-the-art models, *e.g.*, OpenAI-o3 (OpenAI, 2025) scores 54.8%, while setting a new standard for assessing nuanced visual grounding, multi-step reasoning transparency, and cross-modal interaction.

**TreeVGR** advances VGR training through reinforcement learning guided by dual IoU rewards, which explicitly supervise bounding box generation to ensure both precision and recall. This approach enables explainable reasoning pathways and achieves significant improvements across benchmarks.

**Limitation and future works.** The current implementation of **TreeVGR** is based on a 7B parameter model, which may limit scalability compared to larger architectures. **TreeBench** contains only 405 rigorously curated question-answer pairs. Expanding the benchmark with additional samples across broader domains would further challenge model capabilities. Scaling up would be future work.

## ACKNOWLEDGEMENTS

This work was supported by the Beijing Natural Science Foundation (No. L257015) and the National Natural Science Foundation of China (No. 62320106010).

## ETHICS STATEMENT

Our research is grounded in ethical practices, with particular attention paid to the responsible use of data. All datasets employed in this study are publicly available and well-established within the computer vision community. Specifically, our benchmarking was conducted on SA-1B (Kirillov et al., 2023). Our use of this data is in accordance with their provided licenses and intended academic purpose.

## REPRODUCIBILITY STATEMENT

We are committed to ensuring the reproducibility of the research presented in this paper. To this end, comprehensive implementation details for our models and experiments are provided in Appendix E, including the training procedures and all hyperparameters used. Furthermore, upon acceptance of this paper, all source code, datasets, and trained model checkpoints will be made publicly available.

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

APPENDIX

## A  OVERVIEW

Here, we provide a table of contents:

## B  ANNOTATION PIPELINE

**TreeBench** was constructed through a systematic pipeline combining automated sampling, LMM-assisted generation, and three rounds of human validation. The annotation team contains eight human experts in LMMs, including six Ph.D candidates and two senior research scientists.

**1. Image Selection.** A total of 1K images are initially sampled from the SA-1B (Kirillov et al., 2023), with deliberate prioritization of images containing high-density objects (*e.g.*, scenes with overlapping or clustered items), as it offers high-resolution, real-world scenes with a large number of small and varied objects, making it particularly suitable for evaluating visual grounded reasoning. To ensure balanced representation across categories, 100 images are initially allocated per category.

**2.  First Round Quality Control.** The annotation team manually evaluates the relevance and quality of each image for its assigned category. This step is critical for addressing category-specific requirements, *e.g.*, the "Ordering" category necessitates images with visually similar or repetitive objects for practical reasoning tasks. Following this review, 647 images meet the criteria.

**3. Automated Question Generation.** Question-option-answer trios are then generated using two advanced LMMs, *i.e.*, OpenAI-o3 (OpenAI, 2025) and Gemini-2.5-Pro (DeepMind, 2025b), each tasked with producing three diverse, high-quality questions per image. Prompts are designed to emphasize task-specific complexity and visual-semantic alignment.

**4. Second Round Quality Control.** Human experts then manually review all six model-generated questions per image. For each image, annotators selected the most semantically coherent and task-relevant question from the pool of six, prioritizing: (1) alignment with the target subtask, (2) avoidance of trivial or ambiguous object referring, and (3) clarity and unambiguous answerability. If none of the six questions met these criteria, annotators manually constructed a new question. This step ensures that only high-quality, human-vetted questions advance to the next stage.

**5. Difficulty Filtering.** Questions deemed insufficiently challenging are removed through model-based consensus screening. Specifically, any question answered correctly by all four state-of-the-art vision-language models (Qwen2.5-VL-72B (Bai et al., 2025a), InternVL3-78B (Zhu et al., 2025), GPT-4o (OpenAI, 2024a), Gemini-2.5-Flash (DeepMind, 2025a)) was excluded to ensure the benchmark retained meaningful difficulty.

**6. Third Round Quality Control.** The final cross-verification phase engages independent human annotators to cross-validate the accuracy and relevance of each question-option-answer pair. The final dataset comprised 405 rigorously validated questions.

## C  BENCHMARK STATISTICS

**Distribution of Each Subtask.** As demonstrated in Figure 4, **TreeBench** emphasizes advanced reasoning tasks, accounting for 63% of the total subtasks (256 questions), while basic perception-

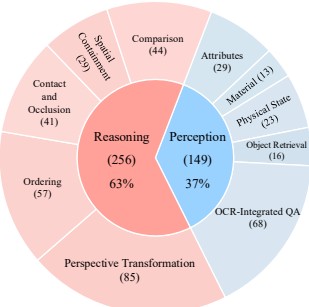

Figure 4: Distribution of each discipline in **TreeBench**, which prioritizes reasoning over perception.

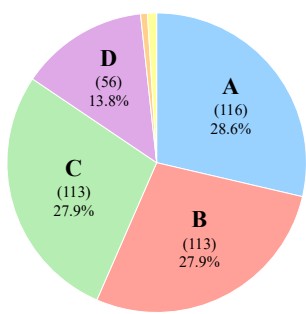

Figure 5: The ground-truth distribution of **TreeBench** with 3 instances of E and 4 instances of F.

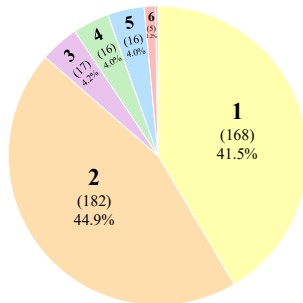

Figure 6: Distribution of the number of instances in **TreeBench**, with one question with 8 target instances.

related tasks constitute 37% (149 questions). Within the reasoning category, key subtasks reflect a focus on complex spatial and relational understanding. This structure underscores a deliberate prioritization of higher-order reasoning over foundational perceptual tasks, aligning with the goal of challenging models to process nuanced relationships and transformations rather than mere object recognition or attribute detection.

**Distribution of Answers.** As illustrated in Figure 5, the ground-truth distribution of **TreeBench** is dominated by four main categories: A (28.6%, 116 instances), B (27.9%, 113 instances), C (27.9%, 113 instances), and D (13.8%, 56 instances). These account for 98.2% of the total 405 instances. The remaining 1.8% (7 instances) includes E (3 instances) and F (4 instances). This structure highlights a balanced emphasis on categories A, B, and C, with D as a notable secondary group, while E and F represent minor but distinct components.

**Distribution of the Number of Target Instances.** Figure 6 shows the distribution of the number of target instances per question. The majority of questions in **TreeBench** require identifying 1 or 2 target instances, accounting for 41.5% (168 questions) and 44.9% (182 questions) of the total, respectively. Questions requiring 3, 4, 5, or 6 targets constitute smaller fractions: 4.2% (17 questions), 4.0% (16 questions), 4.0% (16 questions), and 1.2% (5 questions), respectively. Notably, a single question (highlighted in gray) demands 8 target instances, representing an extreme case. Overall, 86.4% of questions focus on 1–2 targets, suggesting a balance between simplicity and complexity in task design while incorporating rare multi-target scenarios for comprehensive evaluation.

**Distribution of Target Instance Area.** We compute the *relative* area for each target instance using its bounding box, *i.e.*, area $= \frac{1}{HW}(y_2 - y_1)(x_2 - x_1)$, where $H$ and $W$ are the input resolution. Figure 7 is the histogram of the mean area for each question. It illustrates that the majority of target instances in **TreeBench** are extremely small, with a sharp peak near 0.0 and a long tail extending to larger areas (up to 0.7). The mean area across all questions is 0.0305, confirming that targets are predominantly tiny. Most questions (highest frequency bin) involve target instances with areas clustered around 0.0 to 0.05, while only a small fraction require identifying larger objects. This

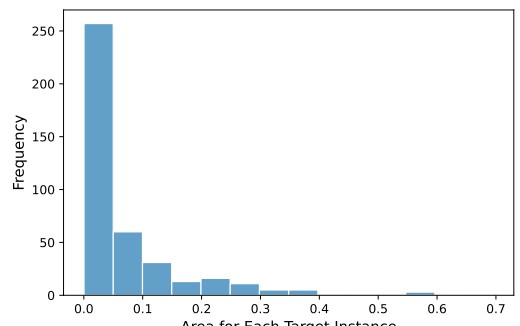

Figure 7: The histogram of mean target instance areas per question with a low average of 0.0305 (indicating small target instances).

distribution highlights the importance of addressing challenging scenarios where small-scale object detection and reasoning are crucial, potentially compromising model performance.

# D ANALYSIS OF TREEBENCH

**Correlation between Localization and Performance.** Importantly, for visual grounded reasoning models, our traceable evaluation demonstrates a *positive correlation* between localization preci-

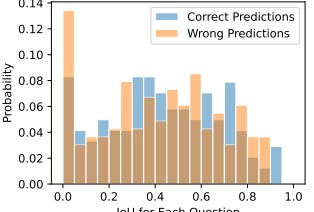

Figure 8: Distribution of IoU for each question in **TreeBench**.

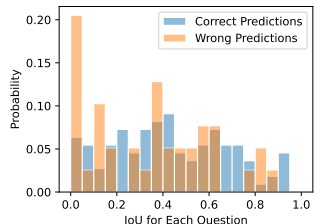

Figure 9: Distribution of IoU for each question in **TreeBench-Perception**.

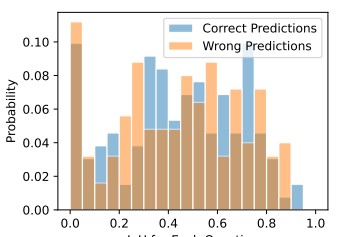

Figure 10: Distribution of IoU for each question in **TreeBench-Reasoning**.

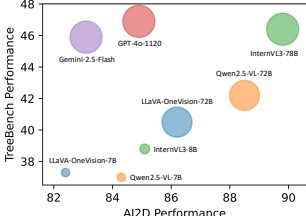

Figure 11: Performance *decoupling* with AI2D (Hiippala et al., 2021).

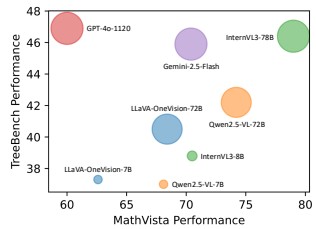

Figure 12: Performance *decoupling* with MathVista (Lu et al., 2023).

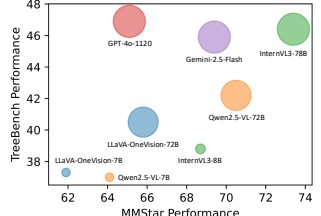

Figure 13: Performance *decoupling* with MMStar (Chen et al., 2024a).

sion and the overall performance, as illustrated in Table 2. This positive correlation between precise localization (mIoU) and overall performance is evident in the progressive improvement from DeepEyes-7B (Zheng et al., 2025b) to Pixel-Reasoner-7B (Su et al., 2025) to our final **TreeVGR-7B**. As mIoU increases, the overall scores rise correspondingly, with **TreeVGR-7B** achieving the highest mIoU and strongest overall performance *at the same time*.

Beyond global analysis, we further plot the histogram of IoU for each question in Figure 8, where blue bars represent wrong predictions and orange bars are correct predictions. Overall, *wrong* predictions tend to have *smaller* IoU values. However, by going deeper through the lens of perception and reasoning, the relationship between mIoU and performance *diverges*. Precise localization (mIoU) aligns closely with perception performance demonstrated in Figure 9. In contrast, as shown in Figure 10, reasoning performance reveals a weaker correlation with mIoU, as improvements in localization alone fail to fully translate to complex reasoning tasks. This disconnect suggests that reasoning questions of **TreeBench** require second-order cognitive capabilities that go *beyond* precise spatial localization.

**Correlation with Other Multimodal Benchmarks.** We systematically compare our **TreeBench** with three existing multimodal benchmarks: AI2D (Hiippala et al., 2021), MathVista (Lu et al., 2023), and MMStar (Chen et al., 2024a), in Figure 11, Figure 12, and Figure 13, respectively, to investigate potential performance correlations. Our analysis reveals a *decoupling* of performance characteristics. For instance, while GPT-4o-1120 (OpenAI, 2024a) ranks among the top performers on **TreeBench**, it lags significantly behind alternatives on other benchmarks. This dissociation underscores the unique emphasis on "thinking with images" of our **TreeBench**.

**The Quality of Visual Evidence in TreeBench.** First, in Table 5, we mask all instances during inference on **TreeBench**. The results show a significant performance drop across all models. This confirms that the bounding boxes in TreeBench are not only high-quality but also indispensable for accurate visual grounded reasoning, directly validating the importance of the annotated evidence. Moreover, we conduct dual experiments in Table 6. When we provide ground-truth bounding boxes as explicit evidence hints to models, all models achieve consistent performance gains. It indicates that bounding boxes indeed help models explicitly anchor their reasoning to visual evidence: without boxes, models may rely on ambiguous textual biases or global image impressions, while with boxes, they are forced to align their answers with the specific visual content in the evidence region, which reflects a shift from "heuristic reasoning" to "evidence-based reasoning".

Table 5: **Performance comparison with *masked* target instances.** When masking out all target instances on **TreeBench**, we observe a significant performance drop across all models, confirming that the annotated bounding boxes are not only high-quality but also indispensable for accurate visual grounded reasoning.

| Masking | Qwen2.5-VL-7B | InternVL3-8B | GPT-4o | o3 | Gemini-2.5-Flash | Gemini-2.5-Pro |
|---------|---------------|--------------|--------|-----|------------------|----------------|
|  | 37.0 | 38.8 | 46.9 | 54.8 | 45.9 | 54.1 |
| ✓ | 31.8 ↓ 5.2 | 29.6 ↓ 9.2 | 29.1 ↓ 17.8 | 33.8 ↓ 21.0 | 29.9 ↓ 16.0 | 33.1 ↓ 21.0 |

Table 6: **Performance comparison with explicit bounding boxes-based textual hints.** When we provide ground-truth bounding boxes as explicit evidence hints to models, all models achieve consistent performance gains.

| Textual Boxes | Qwen2.5-VL-7B | InternVL3-8B | GPT-4o | o3 | Gemini-2.5-Flash | Gemini-2.5-Pro |
|---------------|---------------|--------------|--------|-----|------------------|----------------|
|  | 37.0 | 38.8 | 46.9 | 54.8 | 45.9 | 54.1 |
| ✓ | 43.7 ↑ 6.7 | 43.5 ↑ 4.7 | 49.4 ↑ 2.5 | 58.3 ↑ 3.5 | 51.9 ↑ 6.0 | 61.0 ↑ 6.9 |

# E    IMPLEMENTATION DETAILS

## E.1    COLD-START INITIALIZATION

**Data Construction.** We base our supervised fine-tuning (SFT) dataset on VGR-158K (Wang et al., 2025e), which provides pseudo-chain-of-thought annotations paired with bounding boxes for visual reasoning tasks. However, to align with the grounding capabilities of our base model (Qwen2.5-VL series (Bai et al., 2025a)), which outputs *absolute* coordinates rather than the normalized coordinates (ranging from 0 to 1) used by LLaVA-NeXT (Liu et al., 2024a) in (Wang et al., 2025e), we perform coordinate system conversion. Specifically, for each bounding box, we transform normalized coordinates $[r_{x_1}, r_{y_1}, r_{x_2}, r_{y_2}]$ into *absolute* coordinates via $[x_1, y_1, x_2, y_2] = [Wr_{x_1}, Hr_{y_1}, Wr_{x_2}, Hr_{y_2}]$, where $H \times W$ is the resolution of the input image. Next, we filter samples to prioritize complex reasoning pathways, retaining only entries with multiple bounding boxes (*i.e.*, more than one box per reasoning trajectory). This yields 35K samples, as multi-box interactions demand stronger spatial-temporal reasoning compared to single-box tasks. Subsequently, we construct a reflective subset of 4.7K samples among them by introducing controlled perturbations: for each sample, we (1) inject a synthetic error by inserting a randomly generated incorrect bounding box into the reasoning sequence, and (2) append the meta-cognitive prompt "Wait, this box seems to be wrong" immediately afterward, resulting in our **TreeVGR-SFT-35K**. This design explicitly trains the model to detect and correct erroneous visual grounding, which is a critical skill for robust real-world deployment.

**Optimization.** Initialized from Qwen2.5-VL-7B-Instruct (Bai et al., 2025a), we train **TreeVGR-7B-CI** ("CI" here stands for Cold Initialization) with 8 GPUs using LLaMA-Factory (Zheng et al., 2024), where the AdamW optimizer (Loshchilov & Hutter, 2017) with a learning rate of 5e-6 and a global batch size of 256 is utilized. The learning rate is decayed following a cosine schedule (Loshchilov & Hutter, 2016) with a warmup ratio of 0.1.

## E.2    REINFORCEMENT LEARNING

**Data Construction. TreeVGR** incorporates a novel dual IoU reward, which means each sample should contain ground-truth bounding boxes during the RL phase. To this end, we filter *hard* samples from the original 191K training set of V* (Wu & Xie, 2024) using Qwen2.5-VL-7B-Instruct (Bai et al., 2025a), resulting in 30K samples. Additionally, we incorporate the VisDrone dataset (Zhu et al., 2021), which is originally designed for detection and tracking under UAV images, which offers extremely high-resolution, real-world scenes with a large number of small and varied objects and their corresponding bounding box annotations. We reformulate the training set and the validation set into 38K multiple-choice counting problems, and only retain samples with the ground-truth number ranging from 5 to 10, contributing to the final 7K samples. Finally, our **TreeVGR-RL-37K** consists of 30K open-ended question-answering samples from V* (Wu & Xie, 2024) and 7K multiple-choice problems from VisDrone (Zhu et al., 2021).

**Optimization.** Initialized from **TreeVGR-7B-CI**, we train our final **TreeVGR-7B** with 8 GPUs, with another 8 GPUs serving the reward model, *i.e.*, Qwen2.5-72B-Instruct (Team, 2024), using vLLM (Kwon et al., 2023). We adopt Group Relative Policy Optimization (GRPO) (Shao et al.,

Table 7: Comparison with state-of-the-art alternatives on other multimodal benchmarks, including CV-Bench (Tong et al., 2024a), MMVP (Tong et al., 2024b), MMBench (Liu et al., 2023b), POPE (Li et al., 2023c), AI2D (Hiippala et al., 2021), and ChartQA (Masry et al., 2022). †Results are obtained from (Guo et al., 2025b), otherwise are self-collected.

| Capability | Benchmark | Qwen2.5-VL-7B | TreeVGR-7B | Qwen2.5-VL-72B |
|---|---|---|---|---|
| Vision-centric question answering | CV-Bench-2D | 74.1 | **76.9** ↑ 2.8 | 77.7 |
| | CV-Bench-3D | 72.6 | **77.6** ↑ 5.0 | 87.0 |
| | MMVP | 66.7 | **75.3** ↑ 8.6 | 66.7† |
| General VQA | MMBench$_\text{dev}^\text{en}$ | 83.1 | **84.4** ↑ 1.3 | 88.6† |
| | POPE | 86.7 | **87.2** ↑ 0.5 | 84.9 |
| Document and chart | AI2D$_\text{test}$ | **84.9** | 84.8 ↓ 0.1 | 88.7† |
| | ChartQA$_\text{test}$ | 85.6 | **85.8** ↑ 0.2 | 89.5† |

Table 8: Ablations of each component of our **TreeVGR**. "MME-RW" stands for MME-RealWorld-Lite (Zhang et al., 2024a), and "Acc" represents the multiple-choice accuracy. †This improvement mainly comes from the training set, as many training samples from V* (Wu & Xie, 2024) are included in RL. ‡The model *enumerates* boxes to obtain larger IoU recall, and fails to produce final answers.

| | | Cold-Start | $R_\text{acc} + R_\text{format}$ | $R_\text{IoU}^\text{R}$ | $R_\text{IoU}^\text{P}$ | TreeBench Acc | mIoU | V* Acc | MME-RW Acc |
|---|---|---|---|---|---|---|---|---|---|
| ① | Qwen2.5-VL-7B | | | | | 37.0 | – | 71.2 | 42.3 |
| ② | Cold-Start | ✓ | | | | 39.0 | 23.4 | 76.4 | 48.4 |
| ③ | **TreeVGR** | ✓ | ✓ | ✓ | ✓ | **50.4** | **44.0** | **91.1** | **54.9** |
| ④ | *w/o* Traceable Evidence | ✓ | ✓ | | | 38.0 | 27.2 | 87.9† | 51.6 |
| ⑤ | *w/o* Precision‡ | ✓ | ✓ | ✓ | | 0.0 | 78.3 | 0.0 | 0.0 |
| ⑥ | *w/o* Recall | ✓ | ✓ | | ✓ | 45.4 | 20.6 | 89.5 | 52.6 |
| ⑦ | Text-Only RL | | ✓ | | | 39.0 | – | 86.9† | 46.3 |

2024b), which has been proved to be effective and efficient for diverse tasks. We have also tried DAPO (Yu et al., 2025), but we find it unstable compared with GRPO. Therefore, we simply utilize the original GRPO (Shao et al., 2024b). We implement using EasyR1 (Zheng et al., 2025a), which is a clean fork of veRL (Sheng et al., 2024). We train our **TreeVGR-7B** with 5 epochs on **TreeVGR-RL-37K**, which is significantly less than DeepEyes-7B (Zheng et al., 2025b) (which is trained on 47K samples with *32* epochs).

# F  MORE EXPERIMENTS

## F.1  RESULTS ON OTHER MULTIMODAL BENCHMARKS

In Table 7, we compare our **TreeVGR** with its base model Qwen2.5-VL-7B (Bai et al., 2025a) on a variety of conventional multimodal benchmarks. Specifically, we select CV-Bench (Tong et al., 2024a) and MMVP (Tong et al., 2024b) to evaluate vision-centric question-answering capabilities. MMBench (Liu et al., 2023b) and POPE (Li et al., 2023c) are selected for evaluating general VQA capabilities, and AI2D (Hiippala et al., 2021) and ChartQA (Masry et al., 2022) for comprehension with document and chart. We observe significant improvements in most cases, especially for vision-centric benchmarks. Notably, **TreeVGR-7B** achieves 75.3 on MMVP (Tong et al., 2024b), even surpasses Qwen2.5-VL-72B (Bai et al., 2025a) by a significant margin.

## F.2  ABLATION STUDIES

The core contribution of **TreeVGR** is the *traceable* training pipeline, where the dual IoU reward $R_\text{IoU}$ is incorporated in conventional RL training. Therefore, we aim to evaluate the effectiveness of including this traceable term. As demonstrated in Table 8, we ablate each component of our **TreeVGR**, including the cost-start initialization and reward functions.

**The cold-start stage is quite beneficial for visual grounded reasoning**, when compared with ① and ②. This means the formatting of outputting bounding boxes of target instances is useful for conventional visual grounded reasoning benchmarks like V* Bench (Wu & Xie, 2024) and MME-

RealWorld-Lite (Zhang et al., 2024a). Note that these benchmarks can be regarded as Out-of-Domain (OOD) samples for the SFT dataset.

**Traceable visual grounded reasoning is more effective than untraceable one**, when compared with ③ and ④. Starting from the *same* cold-start checkpoint, integrating dual IoU rewards into the RL framework yields substantial performance gains, particularly on our **TreeBench** and MME-RealWorld-Lite (Zhang et al., 2024a), which represent out-of-distribution (OOD) scenarios relative to the RL training data. Notably, on **TreeBench**, our **TreeVGR** demonstrates significant enhancements in both overall accuracy and mIoU. This dual improvement suggests that precise and interpretable reasoning pathways are critical for achieving optimal performance, underscoring the value of structured reward design in complex, real-world tasks.

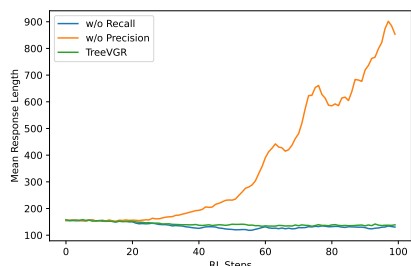

Figure 14: Mean response length with different IoU rewards. The precision term is crucial for alleviating the repetition problem.

**The precision term is crucial for alleviating the repetition problem**, when compared with ③ and ⑤. As illustrated in Figure 14, without precision, the mean response length grows rapidly. When evaluating this model, we find that it tends to *enumerate* candidate bounding boxes to obtain larger IoU recall and thus always fails to produce final answers.

**The recall term is crucial for precise and complete localization**, when compared with ③ and ⑥. On **TreeBench**, without the recall term, the model achieves significant accuracy improvements, but the localization accuracy (mIoU) remains limited, usually grounding *incomplete* target instances.

**Vanilla text-only RL is not so effective as visual grounded reasoning**, when compared with ③ and ⑦. Vanilla RL in text-based tasks demonstrates value through its text-space reasoning capabilities. However, when integrating visual grounded reasoning with traceable evidence, the performance gains become more significant. This highlights the critical role of two factors: (1) pre-answer contextual grounding to anchor responses in multimodal evidence, and (2) accurate spatial localization to refine decision-making precision.

## G    LIMITATIONS AND FUTURE WORKS

One possible limitation of **TreeVGR** is the model scale and architecture, which is limited to Qwen2.5-VL-7B (Bai et al., 2025a). Experiments with other base models and larger model scales could be future work. Furthermore, **TreeVGR** is *not* a general multimodal reasoner, as it is not designed to perform ultra-long reasoning processes in math, sciences, and coding. How to effectively unify vision-centric reasoning models with standard text-centric models could be a future work.

As for **TreeBench**, we find that the "perspective transform" protocol becomes one of the major bottlenecks, which means current state-of-the-art multimodal models, even including visual grounded reasoning models, have *not* effectively modeled the ego-view 3D awareness. Moreover, basic perception capabilities *under complex scenes* are also limited, leading to relatively low scores on "attributes" and "material". How to effectively let LMMs perceive *any* details of the dense visual world becomes a critical challenge.

## H    QUALITATIVE EXAMPLES OF TREEVGR

We first provide qualitative examples of the correlation between the predicted bounding boxes and the model's internal attention maps in Figure 15, where the attention values are the *mean* attention of generated tokens related to bounding boxes (highlighted in red). Strong correlations are observed across examples, indicating that the model's focus is truly guided by those textual bounding boxes-based hints. This explains why visual grounded reasoning pipelines could be such effective. Note that sometimes the visual attention is still *being overwhelmed by textual domain knowledge* even if the grounding is accurate (see the failure case in Figure 15). Explicitly enforcing the model's focus beyond attention may be one potential solution.

Next, we provide qualitative examples and failure cases of our **TreeVGR** on each category of **TreeBench** in Figures 16 to 25.

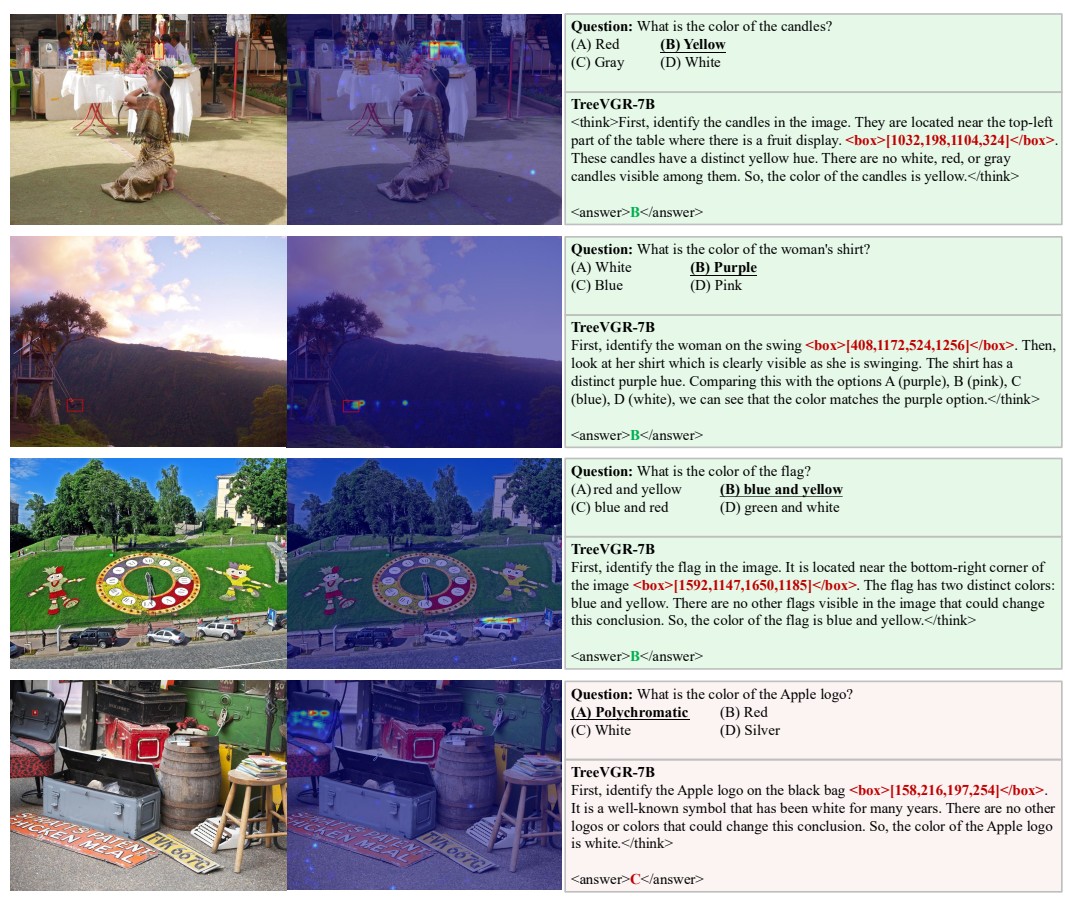

Figure 15: Qualitative examples (first three rows) and failure cases (the last row) on V* Bench (Wu & Xie, 2024) with attention maps.

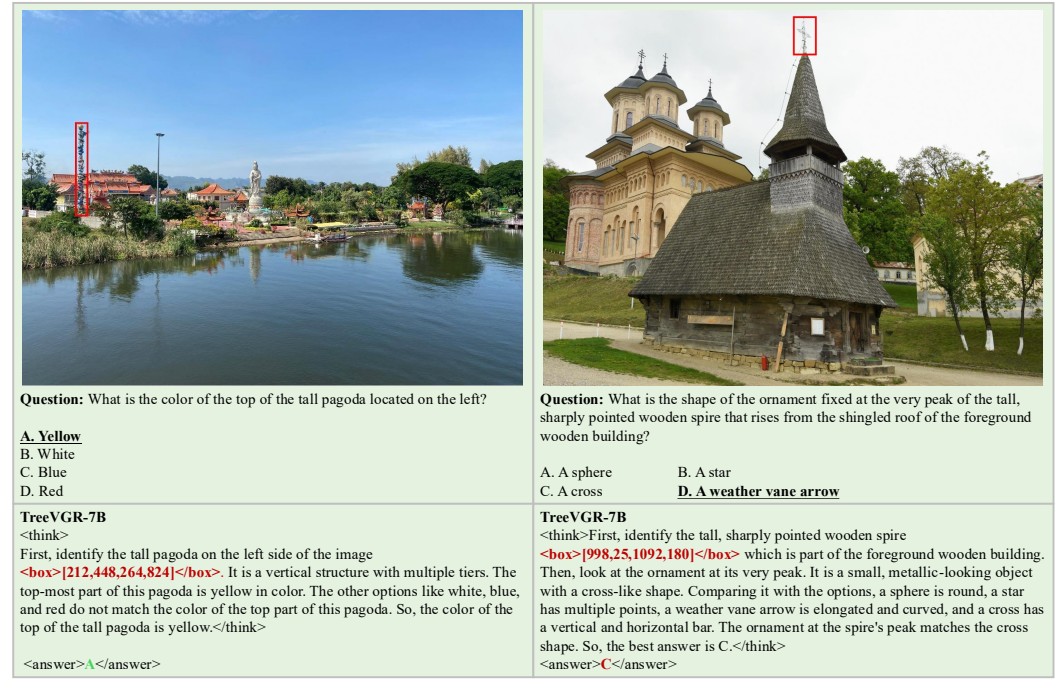

Figure 16: Qualitative examples (left) and failure cases (right) on the "Attributes" protocol of **TreeBench**.

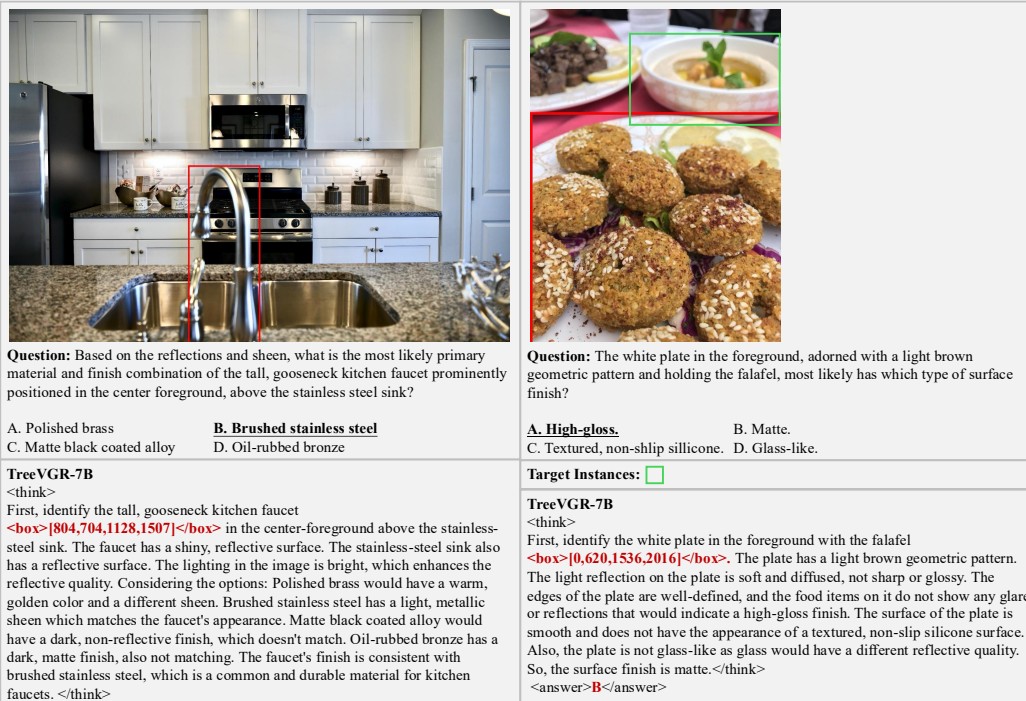

Figure 17: Qualitative examples (left) and failure cases (right) on the "Material" protocol of **TreeBench**.

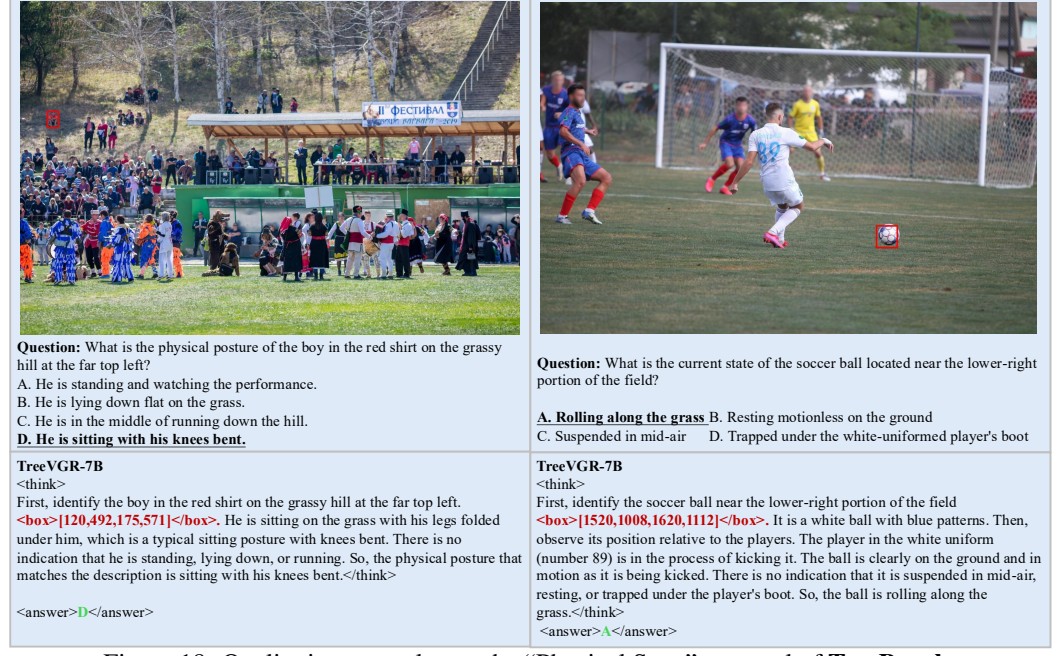

Figure 18: Qualitative examples on the "Physical State" protocol of **TreeBench**.

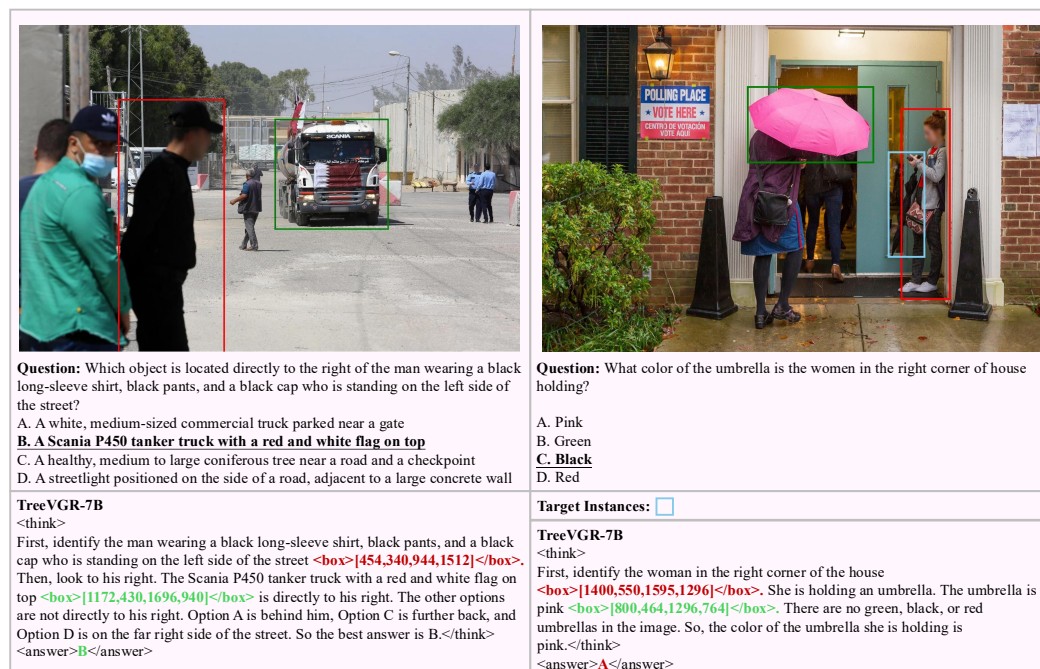

Figure 19: Qualitative examples (left) and failure cases (right) on the "Object Retrieval" protocol of **TreeBench**.

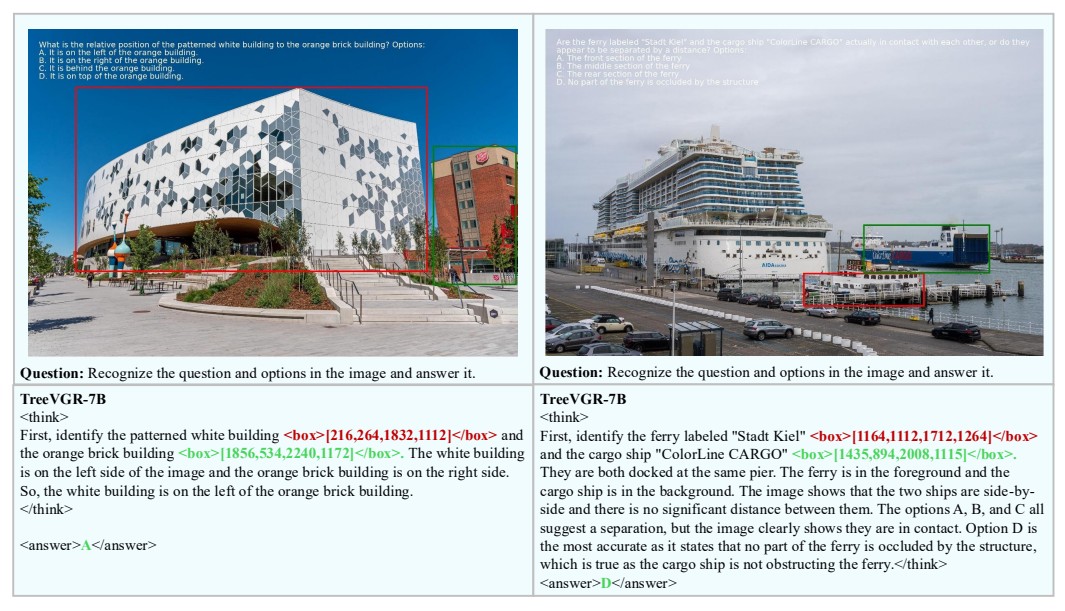

Figure 20: Qualitative examples on the "OCR-Integrated Question-Answering" protocol of **TreeBench**.

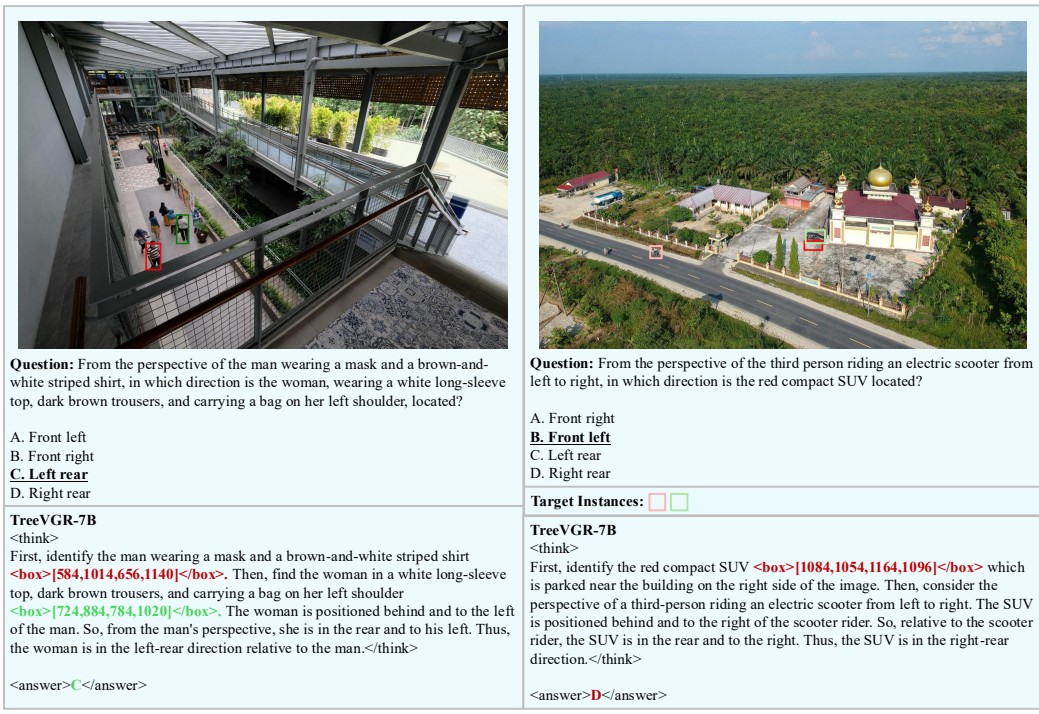

Figure 21: Qualitative examples (left) and failure cases (right) on the "Perspective Transform" protocol of **TreeBench**.

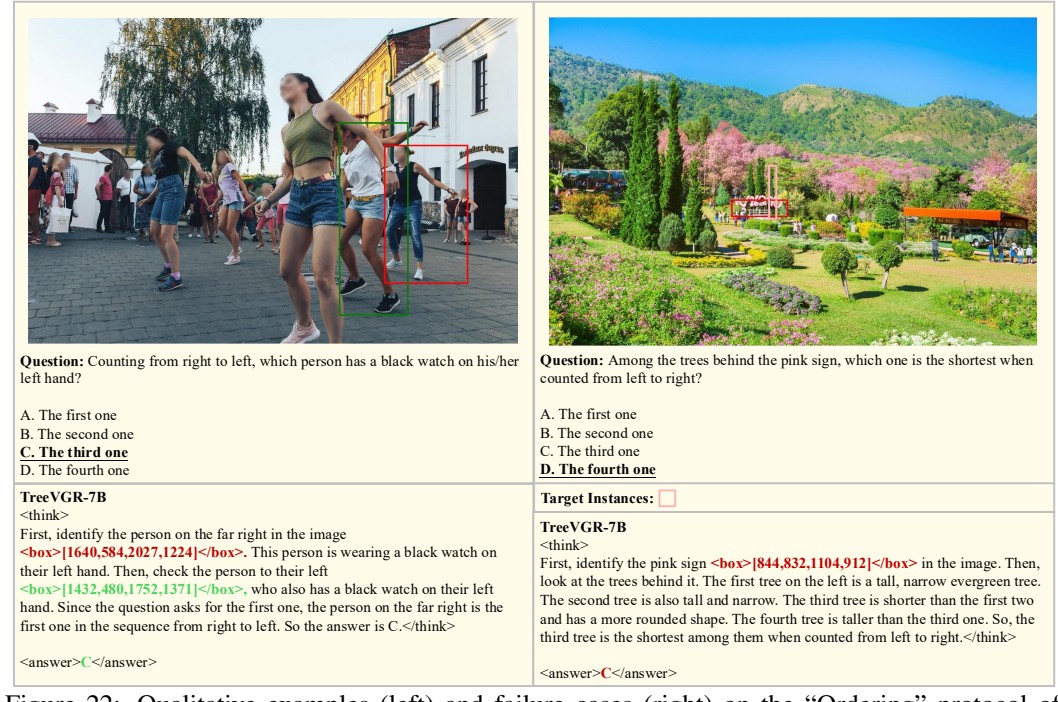

Figure 22: Qualitative examples (left) and failure cases (right) on the "Ordering" protocol of **TreeBench**.

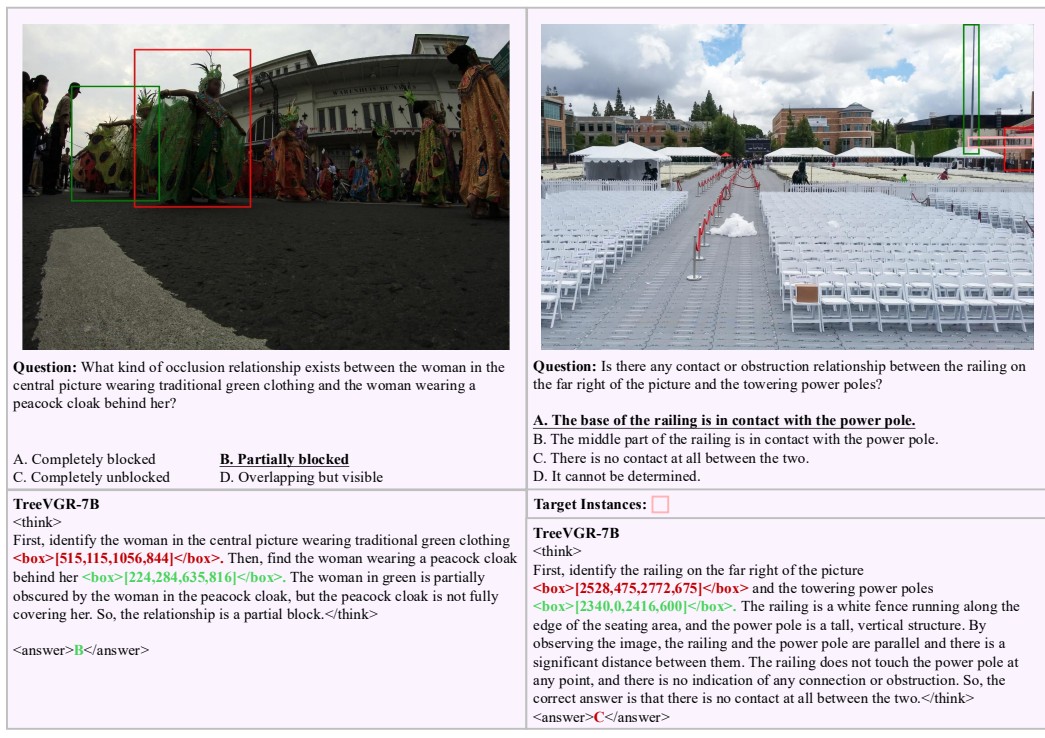

Figure 23: Qualitative examples (left) and failure cases (right) on the "Contact and Occlusion" protocol of **TreeBench**.

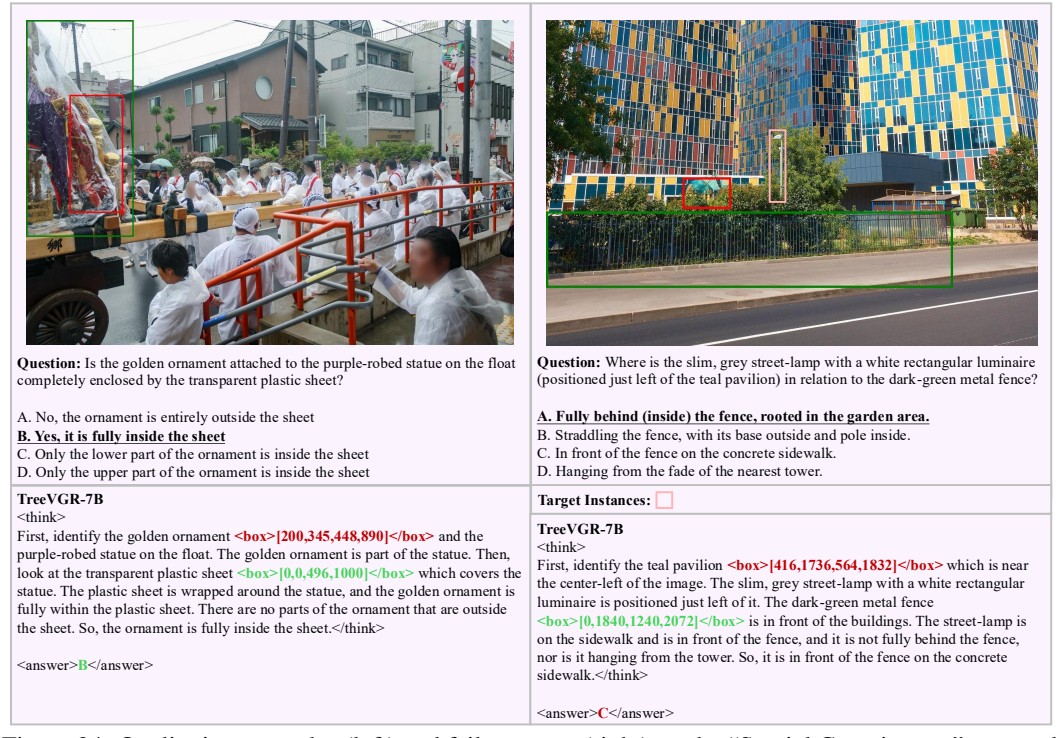

Figure 24: Qualitative examples (left) and failure cases (right) on the "Spatial Containment" protocol of **TreeBench**.

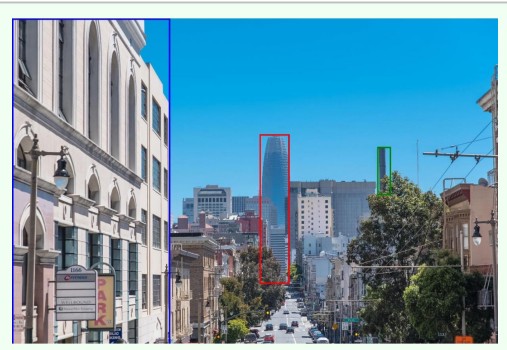 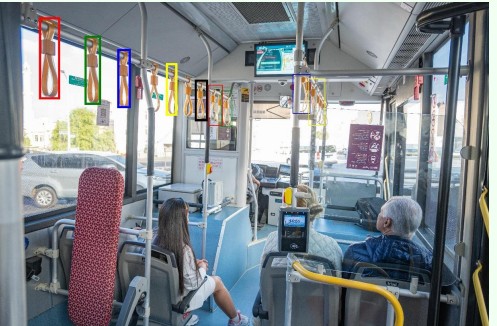

**Question:** In the distant skyline, the actual height of the slender, blue-glass, cone-shaped skyscraper towering in the center of the frame, the narrow, dark gray chimney not far to its right, and the milky white building to the left of the frame. Which object in the photo is actually taller?

A. The smokestack appears tallest among the 3 buildings
B. Themilky white buildin appears tallest among the 3 buildings
**C. The blue-glass skyscraper appears stallest among the 3 buildings**
D. All objects are not visible clearly enough to judge height.

**TreeVGR-7B**
<think>
First, identify the three key buildings in the image. The slender, blue-glass, cone-shaped skyscraper **<box>[1144,534,1283,1224]</box>** is the tallest among the three. The narrow, dark gray chimney **<box>[1688,592,1752,816]</box>** is the shortest. The milky white building **<box>[0,0,731,1512]</box>** is in the left-hand side of the frame and is shorter than the skyscraper. So, the skyscraper is the tallest among the three.</think>

<answer>C</answer>

**Question:** In the picture, the left side of the bus extends from the back of the car to the front of the car, a total of 10 hand straps, counting from 1 to 10, which is the actual distance and the little girl's head is the closest?

A. 2          B. 6
**C. 4**          D. 8

**TreeVGR-7B**
<think>
First, identify the left-side hand straps in the image.
**<box>[192,75,288,450]</box>** <box>[400,160,475,475]</box>
**<box>[550,220,611,490]</box>** <box>[772,285,824,525]</box>
**<box>[904,360,965,550]</box>** <box>[965,385,1032,570]</box>
<box>[1024,410,1075,575]</box> <box>[1352,335,1432,515]</box>
<box>[1424,355,1500,570]</box> Counting these, there are 10 hand straps. The little girl's head is the closest among them. So, the left-side of the bus from the back of the car to the front of the car is 10 units long, and the little girl's head is the closest. The question asks for the option that matches this distance and the position of the little girl's head. The option 6 is the closest match as it is the closest among the given options. </think>

<answer>B</answer>

Figure 25: Qualitative examples (left) and failure cases (right) on the "Comparison" protocol of **TreeBench**.

