# OpenReview forum: "Traceable Evidence Enhanced Visual Grounded Reasoning: Evaluation and Method"
_ICLR.cc/2026/Conference — ICLR 2026 Poster_

### Official Review · Reviewer_o6mU · 2025-10-31

**Soundness:** 3
**Presentation:** 2
**Contribution:** 2
**Rating:** 4
**Confidence:** 4

**Summary:**

This paper aims to assess and improve the grounding reasoning of Vision-Language Models. To this end, it initially introduces a small, high-difficulty benchmark targeting detailed visual queries. Furthermore, the authors present a two-stage training method, which is demonstrated to boost the performance of Qwen2.5-VL-7B on their self-constructed benchmark.

**Strengths:**

1. The proposed benchmark fills a gap in existing open-source benchmarks by specifically testing a model's ability to attend to fine-grained details
2. The proposed method significantly improves the performance of Qwen2.5-VL-7B on their self-constructed benchmark

**Weaknesses:**

1. Regarding the ground truth for the intermediate reasoning: 1) How was this data obtained or generated? 2) What measures were taken to ensure its correctness and accuracy? 3) What was the underlying method and rationale for its construction?
	2. The authors have collected a 37k-sample dataset for RL, but it is unclear how the contributions of the data and the proposed method are disentangled. The paper would be strengthened by an ablation study that isolates the impact of each component. To provide a clearer picture of the method's efficacy, I suggest including a baseline that applies conventional RL algorithms to the same 37k dataset. This would help clarify whether the performance gains stem from the novel method itself or the curated data.
	3. The paper would benefit from a clearer justification for the evaluation dimensions presented in Section 3. Could the authors elaborate on the rationale behind their selection and explain the method used to ensure their comprehensiveness? Furthermore, providing data distribution across these different dimensions would be crucial.
	4. The paper only provides results on a self-constructed benchmark that focuses on small regions. Provide results on public benchmarks. Besides, does this specialized training approach compromise the model's generalization ability?
	5. Does focusing on a single small point per image create bias? This approach may allow the model to succeed without understanding the global context of the image.

**Questions:**

Please refer to the issues detailed in the Weakness part.

---

> ### Author Response · Authors · 2025-11-19
> **Rebuttal to Reviewer o6mU (Part 1)**
>
> We thank Reviewer o6mU for your time and valuable feedback. We are glad you recognized that our proposed benchmark "**fills a gap**" in the field by testing a model's ability to attend to fine-grained details. We also appreciate your acknowledgment that our proposed method "**significantly improves the performance**" of the base model on our self-constructed benchmark.
>
> **W1: About the ground-truth of intermediate reasoning.**
>
> **A1**: The intermediate reasoning contains (1) bounding boxes and (2) text reasoning traces. These signals vary across three key datasets (TreeBench test set, SFT training set, RL training set) due to their different roles. We elaborate on their generation, correctness guarantees, and design rationale in detail below:
>
> (1) First, for the testing set (TreeBench), we only annotate bounding boxes, and no text reasoning traces are used, **as reported in Appendix B in the original manuscript**:
>
> - How it was obtained: Bounding boxes of target instances (e.g., small objects, interacting objects) are manually annotated by 8 LMM experts. Each expert received training on "fine-grained object localization" (e.g., distinguishing similar distractors in dense scenes) to ensure annotation consistency.
> - Measures for correctness: We implement a 3-round quality control pipeline (detailed in Appendix B) with a dedicated cross-validation stage.
> - Rationale for design: We choose manual annotation over model generation because:
>   - TreeBench focuses on "subtle targets in complex scenes" (e.g., 3.05% average target area, Figure 7). Models (even OpenAI-o3) often fail to localize such small objects accurately, leading to noisy ground truth.
>   - Human experts can integrate contextual logic that models currently lack, ensuring alignment with "second-order reasoning" evaluation goals (e.g., occlusion, perspective transformation).
>
> (2) Next, for the SFT set, we actually sample a subset of VGR [1], **which is reported in Appendix E.1 in our original version**. We carefully check the paper and try to answer your question.
> - How it was obtained: The bounding boxes and text traces are both generated by powerful MLLMs.
> - Measures for correctness: To ensure its correctness, they perform rejection sampling and some filter techniques, such as judging whether the cropped images using bounding boxes are relevant to solving the question or not.
> - Rationale for design: The method is to first leverage MLLMs to produce bounding boxes of relevant objects, and then perform rejection sampling and filter strategies. Finally, another MLLM is used to merge them into fluent reasoning traces interleaved with text and bounding boxes.
>
> (3) The RL set focuses on refining spatial grounding precision (via dual IoU reward), so its ground truth only needs high-quality bounding boxes (logical reasoning is already initialized in SFT). We reused and filtered existing public datasets to ensure domain relevance, **which is reported in Appendix E.2 in our original manuscript**.
> - How it was obtained:
>   - 30K samples from V*: We filtered "hard samples" using Qwen2.5-VL-7B to focus on challenging spatial tasks. Bounding boxes are from its official annotation.
>   - 7K samples from VisDrone: A UAV dataset with dense small objects (consistent with TreeBench's "small target" focus). We reformulated its detection annotations into "counting questions" and retained samples with 5 to 10 targets.
> - Measures for correctness: We simply regard those official bounding boxes as correct.
> - Rationale for design:
>   - Reusing public datasets (V*/VisDrone) avoids "data leakage" risks (these datasets are unrelated to TreeBench) and ensures ground truth is domain-recognized (no need for redundant re-annotation).
>   - The "hard sample filtering" step aligns the RL set with our goal of improving "difficulty-aware grounding". Samples that are easy for the base model do not contribute to RL refinement.

---

> ### Author Response · Authors · 2025-11-19
> **Rebuttal to Reviewer o6mU (Part 2)**
>
> **W2: About the contribution of each component.**
>
> **A2**: We have explicitly addressed this via ablation studies in Table 6 by controlling for the "same 37k dataset" while varying only the "method components" (i.e., whether to include TreeVGR's core design: dual IoU reward).
> When comparing the 3rd row (our TreeVGR) and the 4th row (conventional GRPO on the same 37k dataset), integrating dual IoU rewards into the RL framework yields substantial performance gains, particularly on our TreeBench and MME-RealWorld-Lite, which represent out-of-distribution (OOD) scenarios relative to the RL training data. Notably, on TreeBench, our TreeVGR demonstrates significant enhancements in both overall accuracy and mIoU. This dual improvement suggests that precise and interpretable reasoning pathways are critical for achieving optimal performance, underscoring the value of structured reward design in complex, real-world tasks.
>
> | Metric | 4th Row (Conventional GRPO on 37K Data) | 3rd Row (TreeVGR on 37K Data) |
> |-|-|-|
> | TreeBench Accuracy     | 38.0 | 50.4 (+12.4) |
> | TreeBench mIoU         | 27.2 | 44.0 (+16.8) |
> | MME-RealWorld-Lite Acc | 51.6 | 54.9 (+3.3) |
> | V* Bench Accuracy      | 87.9 | 91.1 (+3.2) |
>
> *Note that the above results are pasted from Table 6 in our original manuscript for clarity.*
>
> **W3: About the evaluation dimensions on TreeBench.**
>
> **A3**: Thank you for requesting clarity on the evaluation dimensions in Section 3. The 10 core competencies (5 for Perception, 5 for Reasoning) were *not* arbitrarily selected. They were designed to address critical gaps in existing VGR benchmarks (Section 2 Related Work) and cover the full pipeline of "thinking with images" (from basic visual grounding to advanced logical inference). We elaborate on their rationale, comprehensiveness guarantees, and detailed data distribution below.
>
> (1) Rationale: The underlying rationale behind this selection mainly includes (i) the ability to identify small target objects with long, detailed, and unique text captions in large, complex, and real-world scenes, and (ii) second-order reasoning beyond precise localization.
>
> (2) Comprehensiveness: To avoid missing key VGR capabilities, the 10 dimensions map to the human "thinking with images" process. (i) First, "Perception" (5 dimensions) ensures the model can "see correctly" (e.g., locate small objects, recognize attributes), which is the foundation of grounded reasoning. (2) Then, "Reasoning" (5 dimensions) tests the model’s ability to "infer beyond seeing" (e.g., perspective shifts, object interactions), which is the core of VGR that differentiates it from simple object detection. We have tried our best to be as comprehensive as possible, as we select 10 distinct and important aspects. We acknowledge that there might be other aspects, as comprehensiveness can hardly be ensured.
>
> (3) Data Distribution: **We have already provided the distribution of each discipline in TreeBench in Figure 4 in our original manuscript.** TreeBench emphasizes advanced reasoning tasks, accounting for 63% of the total subtasks (256 questions), while basic perception-related tasks constitute 37% (149 questions). Within the reasoning category, key subtasks reflect a focus on complex spatial and relational understanding. This structure underscores a deliberate prioritization of higher-order reasoning over foundational perceptual tasks, aligning with the goal of challenging models to process nuanced relationships and transformations rather than mere object recognition or attribute detection.
>
> **W4: Performance on public benchmarks and general capabilities.**
>
> **A4**: In **Tables 3 and 4 in our original manuscript**, we have provided comparisons on public benchmarks like V*, MME-RealWorld-Lite, and HR-Bench-4K, HR-Bench-8K, where TreeVGR brings significant improvements over baselines and outperforms other open-source visual grounded reasoning alternatives (DeepEyes and Pixel-Reasoner). Moreover, **Table 5 in our original manuscript** demonstrates that TreeVGR maintains the general capabilities, while it even brings improvements on vision-centric benchmarks, implying its strong generalization capabilities.
>
> **W5: About understanding the global context.**
>
> **A5**: We acknowledge that there are a part of the subtasks of TreeBench that focus on small points, such as "attributes" and "material". However, *there are more categories that require necessary global contexts* such as reasoning questions. As demonstrated in Figure 4, TreeBench emphasizes advanced reasoning tasks, accounting for 63% of the total subtasks (256 questions), while basic perception-related tasks constitute 37% (149 questions). Moreover, from another perspective shown in Figure 6, only 41.5% of TreeBench questions (168/405) require identifying 1 target, while the other 58.5% (237/405) require 2+ targets, which inherently demand global context to resolve relationships between targets.
>
> [1] VGR: Visual Grounded Reasoning. arXiv 2025.

---

### Official Review · Reviewer_qSgY · 2025-10-31

**Soundness:** 2
**Presentation:** 2
**Contribution:** 2
**Rating:** 4
**Confidence:** 5

**Summary:**

This paper tackles the challenge of making vision language models (VLMs) not only answer visual questions correctly but also show where in the image their reasoning comes from. Current VLMs like Qwen2.5-VL or GPT-4V often produce correct answers without verifiable visual grounding, leading to untraceable or hallucinated reasoning.

To address this, the authors propose TreeVGR, a two-stage training framework built on Qwen2.5-VL-7B, and introduce TreeBench, a new benchmark for evaluating traceable visual grounded reasoning.

TreeBench contains manually verified image–question–answer triplets with annotated bounding boxes marking the visual evidence, covering ten sub-tasks spanning perception (e.g., color, attribute, OCR) and reasoning (e.g., ordering, contact, spatial containment).

TreeVGR first performs supervised fine-tuning to teach the model to produce structured reasoning with bounding boxes, then applies reinforcement learning with a reward that combines answer correctness, reasoning format, and bounding-box IoU.

The resulting TreeVGR-7B achieves nice gains over the base Qwen2.5-VL-7B on TreeBench (+13.4 points) and shows moderate improvements on other multimodal reasoning benchmarks such as V*Bench and MME-RealWorld. The authors claim that TreeVGR enables more transparent and verifiable visual reasoning by linking model predictions to explicit image evidence.

**Strengths:**

1. TreeBench is carefully constructed and manually verified for correctness and visual traceability, providing one of the first benchmarks that explicitly links reasoning answers to bounding-box evidence.

2. The paper identifies a gap in current multimodal research: the lack of verifiable, evidence-grounded reasoning, and frames the need for “traceable visual reasoning” in a straightforward way.

3. The two-stage TreeVGR pipeline (supervised fine-tuning followed by RL with evidence-based rewards) is simple, reproducible, and effectively demonstrates that incorporating bounding-box supervision can improve visual reasoning performance.

4. The experiments cover diverse perception and reasoning sub-tasks, results are consistently reported, and the writing is clear, making both the dataset and the approach easy to understand and potentially useful for future follow-up work.

**Weaknesses:**

1. TreeBench, though high quality, relies heavily on manual verification of image–question–evidence triplets, making it difficult to scale to larger or more diverse data. The process is partly automated but still human-dependent, which restricts reproducibility and extensibility.

2. The paper equates correct reasoning with the ability to predict accurate bounding boxes, but provides no empirical evidence that models failing to output boxes are not attending to the correct regions internally. This makes the “traceability” assumption more procedural than cognitive, and potentially misleading.

3. The model is trained and tested on tasks with nearly identical structures and output formats (question + bounding-box evidence). As a result, the large reported gains likely reflect adaptation to the curated data and benchmark design rather than a general improvement in reasoning ability.

4. The RL stage yields only minor improvements over supervised fine-tuning and is insufficiently analysed. The paper does not demonstrate that reinforcement learning meaningfully enhances reasoning depth or grounding beyond improving output formatting.

5. The study evaluates only on reasoning-style benchmarks and does not test whether TreeVGR retains the broader multimodal skills of the base Qwen2.5-VL model. This leaves open the possibility of overfitting or catastrophic forgetting of non-reasoning capabilities.

6. Although the method claims to enhance traceable reasoning, the paper never validates whether the predicted evidence regions align with the model’s internal attention patterns or decision process, limiting the interpretability claims it makes.

**Questions:**

1. How do the authors envision scaling TreeBench to larger or more diverse datasets without compromising annotation quality or requiring extensive human effort, since this is one of the reasons the authors claim their dataset to be superior?

2. The paper assumes that correct reasoning must manifest through accurate bounding-box prediction, yet models may still attend to the correct region internally without explicitly outputting coordinates. Have the authors analysed attention maps or other internal signals to verify that bounding-box accuracy genuinely reflects visual grounding?

3. Since both the training data and TreeBench share nearly identical QA structures and output formats, to what extent do the reported gains represent task adaptation rather than generalizable reasoning improvements? Have the authors tested transfer to unseen reasoning styles or datasets?

4. The RL component seems to add only marginal improvements. Can the authors clarify what qualitative or behavioural differences emerge after RL fine-tuning compared to supervised fine-tuning alone, and whether these differences justify the added complexity?

5. Given that TreeVGR fine-tunes the full Qwen2.5-VL model, have the authors evaluated whether general tasks such as captioning or VQA are preserved, or does the model overfit to the traceable reasoning format?

6. The paper claims that the method enhances “traceable reasoning,” but without analyzing the alignment between predicted evidence and model attention. Can the authors provide evidence—such as attention heatmaps or token-level visualization—that the model truly focuses on the localized regions it predicts or what was the before vs after effect of their method on the base model?

---

> ### Author Response · Authors · 2025-11-19
> **Rebuttal to Reviewer qSgY (Part 1)**
>
> We are very grateful to Reviewer qSgY for your thorough review and encouraging feedback. We are delighted that you found our work to "**identify a gap**" in current research and frame the problem in a "**straightforward way**". We also appreciate your positive assessment of TreeBench as a "**carefully constructed**" and "**manually verified**" benchmark. Furthermore, we are encouraged by your characterization of our TreeVGR pipeline as "**simple, reproducible, and effective**". Finally, we are very thankful for your kind words on the clarity of our writing and presentation, which you noted makes our work "**easy to understand and potentially useful for future follow-up work**",
>
> **W1: About the scalability of TreeBench.**
>
> **A1**: We argue that the most critical aspects of a modern reasoning benchmark are **(1) difficulty and diversity** to truly challenge models, and **(2) correctness** with traceability to ensure meaningful evaluation. This **"less is more"** principle is powerfully demonstrated in recent works like [1, 2, 3, 4], which argue that many large-scale benchmarks are diluted with "overly simple or uninformative samples", making it difficult to distinguish model performance. While we acknowledge the importance of scalability, we believe it should not be pursued at the expense of these foundational pillars.
>
> Specifically, LIME [1] achieves a 76% reduction from original datasets via a semi-automated pipeline that filters uninformative items and eliminates answer leakage. It outperforms larger benchmarks in differentiating model abilities while cutting evaluation time by 77%. Metabench [2] further validates this by distilling six prominent LLMs benchmarks (ARC, GSM8K, MMLU, etc.) into a sparse set comprising less than 3% of the original items, achieving accurate reconstruction of individual and total benchmark scores with a root mean square error (RMSE) as low as 0.58%. Pacchiardi et al. [3] complement this by showing that merely 100 reference instances are sufficient to predict a new LLM's performance on unseen tasks, with results comparable to assessors trained on full datasets. This highlights that strategic sample selection outweighs sheer quantity. Even in the context of model unlearning, Krishnan et al. [4] indirectly support this principle: their findings that frequently encountered data is harder to unlearn underscore that sample quality (e.g., frequency, relevance) has a more profound impact on model behavior and evaluation outcomes than sheer volume. Collectively, these works confirm that bloated benchmarks with redundant or low-value samples not only waste computational resources but also obscure meaningful performance differences, reinforcing that **"more" samples do not equate to "better" evaluation.**
>
> Similar to [1, 2, 3, 4], TreeBench's primary goal is to fill the gap of "traceable visual grounded reasoning" in existing benchmarks. **For a *diagnostic benchmark*** (rather than a large-scale training dataset), **we prioritize annotation correctness and evidence traceability**, as only high-quality, human-verified triplets (image-question-bounding box) can reliably test whether models truly “think with images”. This is why we involved 8 LMM experts (6 PhD candidates + 2 senior researchers) and implemented three rounds of quality control (detailed in Appendix B). These steps are not arbitrary but are intended to guarantee TreeBench's ability to diagnose model weaknesses in traceable reasoning.
>
> Second, regarding reproducibility, we have documented the entire annotation process in extreme detail in Appendix B, including the selection criteria of 8 LMM experts (e.g., 2+ years of multimodal research experience), the three-stage quality control checklist (e.g., criteria for rejecting LMM-generated questions). This ensures that other researchers can fully replicate our annotation workflow to extend TreeBench or verify its quality.
>
> In summary, TreeBench's initial focus on human verification is to establish a high-quality standard for traceable reasoning evaluation, which has been adopted by current representative models like GLM-4.5-V [5] and DeepEyesV2 [6]. Addressing scalability without compromising its core value of correctness could be our future work.

---

> ### Author Response · Authors · 2025-11-19
> **Rebuttal to Reviewer qSgY (Part 2)**
>
> **W2: About the "traceability" assumption.**
>
> **A2**: We never intend to deny that models *might* attend to correct regions internally without explicit box output. However, **internal attention is usually untraceable and unverifiable**, while explicit bounding boxes serve as a critical bridge to link implicit cognitive processes with explicit, verifiable evidence.
>
> **(1) “Explicit bounding box output” correlates with more reliable cognitive evidence**, while internal attention is difficult to directly interpret or validate at scale. As shown in the following table, when we provide ground-truth bounding boxes as explicit evidence hints to models, all models achieve consistent performance gains. This improvement cannot be explained by “procedural formatting”. Instead, it indicates that bounding boxes help models explicitly anchor their reasoning to visual evidence: without boxes, models may rely on ambiguous textual biases or global image impressions, while with boxes, they are forced to align their answers with the specific visual content in the evidence region, which reflects a shift from “heuristic reasoning” to “evidence-based reasoning”.
>
> | Input | Qwen2.5-VL-7B | InternVL3-8B | GPT-4o | o3   | Gemini-2.5-Flash | Gemini-2.5-Pro |
> |-|-|-|-|-|-|-|
> | Question  | 37.0 | 38.8 | 46.9   | 54.8 | 45.9 | 54.1 |
> | Question + Textual Boxes | 43.7 | 43.5 | 49.4   | 58.3 | 51.9 | 61.0 |
>
> **(2) "Accurate bounding boxes" are not just a "procedural output", and the localization accuracy is correlated with the final reasoning performance.** We have validated the correlation between “box accuracy (mIoU)” and “reasoning performance” in TreeVGR (**Table 6 in our original manuscript**). We paste the core experiments in the following.
>
> | ID in Table 6 | Method | Cold-Start | $R_{acc} + R_{format}$ | $R_{IoU}$ | TreeBench-Acc | TreeBench-IoU | V*   | MME-RealWorld-Lite |
> |-|-|-|-|-|-|-|-|-|
> | 3 | TreeVGR | Yes | Yes | Yes | 50.4 | 44.0 | 91.1 | 54.9 |
> | 4 | w/o Traceable Evidence | Yes | Yes | No | 38.0 | 27.2 | 87.9 | 51.6 |
>
> As demonstrated in the table, not supervising bounding boxes leads to significant drops in both final accuracies and localization precisions. The positive correlation between box precision and final accuracy demonstrates that “accurate bounding boxes” are not just a “procedural output”. They correspond to a more precise localization of evidence, which directly translates to better reasoning quality. If boxes were merely a formal requirement, we would not observe such a strong link between mIoU and overall performance.
>
> To be honest, we acknowledge that “explicit box output” is not the only way to reflect internal attention, but it is currently **one of the most verifiable ways**. For example, a model may claim "the girl wears a white skirt" without a box, but how do we confirm that it did not hallucinate? TreeBench's bounding box annotations and TreeVGR's box output solve this by forcing the model to "show its work". The box's alignment with ground truth becomes a measurable metric for whether the model's reasoning is rooted in real visual evidence, rather than textual bias or guesswork. This is the core value of "traceability", not to deny internal attention, but to make that attention *verifiable* and *accountable*.

---

> ### Author Response · Authors · 2025-11-19
> **Rebuttal to Reviewer qSgY (Part 3)**
>
> **W3: The model is trained and tested on tasks with nearly identical structures and output formats, and thus, the gains do not come from general improvement in reasoning ability.**
>
> **A3**: *Disagree*. TreeVGR's improvements stem from enhanced reasoning ability rather than data/format adaptation.
>
> First, we explicitly **decouple the training and test data/task structures** to eliminate "adaptation to curated design". TreeBench's images are sampled exclusively from SA-1B. In contrast, TreeVGR's training data has zero overlap with SA-1B: the RL phase uses 30K samples from V* and 7K samples from VisDrone. The SFT phase uses 35K samples from VGR-158K, which also does not include SA-1B images. This strict image source separation rules out "data leakage or scene adaptation" as a cause of gains on TreeBench.
>
> Second, **TreeVGR's improvements extend to other public benchmarks**, like V*, MME-RealWorld-Lite, and HR-Bench, further confirming general reasoning ability.
>
> *We paste related results from Tables 3 and 4 in our original manuscript in the following for better clarity.*
>
> | Method | V* Bench | HR-Bench-4K | HR-Bench-8K | MME-RealWorld-Lite |
> |-|-|-|-|-|
> | Qwen2.5-VL-7B | 74.3 | 72.1 | 55.5 | 42.3 |
> | TreeVGR-7B | 91.1 (+16.8) | 77.1 (+5.0) | 64.0 (+8.5) | 54.9 (+12.6) |
>
> Third, the "output format (box + answer)" is not a "task-specific design" but a **general paradigm for explicit evidence grounding that transfers to any fine-grained multimodal task**, which is a core contribution precisely because of its universality. This format forces models to "anchor answers to visual evidence". For example (Table 5 in the original manuscript), on CV-Bench, a benchmark for low-resolution object attribute recognition, TreeVGR still improves by +2.8 on CV-Bench-2D and +5.0 on CV-Bench-3D. This confirms the format's value lies in teaching models "evidence-based reasoning", not adapting to a specific task.
>
> *We paste related results from Table 5 in our original manuscript in the following for better clarity.*
>
> | Method | CV-Bench-2D | CV-Bench-3D | MMVP | MMBench | POPE | AI2D | ChartQA |
> |-|-|-|-|-|-|-|-|
> | Qwen2.5-VL-7B | 74.1 | 72.1 | 66.7 | 83.1 | 86.7 | 84.9 | 85.6 |
> | TreeVGR-7B | 76.9 (+2.8) | 77.6 (+5.0) | 75.3 (+8.6) | 84.4 (+1.3) | 87.2 (+0.5) | 84.8 (-0.1) | 85.8 (+0.2) |
>
> **W4: The RL stage yields minor improvements.**
>
> **A4**: *Disagree*. **(1) The RL stage brings significant improvements**. As demonstrated in **Table 6 in the original manuscript**, when comparing the 2nd row (the SFT baseline) and the 3rd row (our TreeVGR), the RL stage brings +11.4 on TreeBench, +14.7 on V*, and +6.5 on MME-RealWorld-Lite, which are definitely significant.
>
> **(2) The RL stage not only improves output formatting but also improves localization precision.** When comparing the 3rd row (our TreeVGR) and the 4th row (w/o box supervision in RL), with box supervision, the RL stage significantly improves localization precision (+16.8 mIoU on TreeBench), contributing to improved performances across benchmarks. This indicates that, beyond simply following the output format alone, precise localization is also important. This dual improvement suggests that precise reasoning pathways are critical for achieving optimal performance, underscoring the value of structured reward design in complex, real-world tasks.
>
> *We simply paste these results for better clarity. All these experiments can be found in Table 6 in our original manuscript.*
>
> | ID in Table 6 | Method | Cold-Start | $R_{acc} + R_{format}$ | $R_{IoU}$ | TreeBench-Acc | TreeBench-IoU | V* | MME-RealWorld-Lite |
> |-|------------------------|------------|------------------------|-----------|---------------|---------------|------|--------------------|
> | 2             | SFT Baseline           | Yes        | --                     | --        | 39.0          | 23.4          | 76.4 | 48.4               |
> | 3             | TreeVGR                | Yes        | Yes                    | Yes       | 50.4          | 44.0          | 91.1 | 54.9               |
> | 4             | w/o Traceable Evidence | Yes        | Yes                    | --        | 38.0          | 27.2          | 87.9 | 51.6               |
>
> **W5: About broader multimodal skills.**
>
> **A5**: As demonstrated in **Table 5 in the original manuscript**, TreeVGR maintains performance on general benchmarks and even brings significant improvements in vision-centric benchmarks. Please refer to **A3** for detailed numbers.

---

> ### Author Response · Authors · 2025-11-19
> **Rebuttal to Reviewer qSgY (Part 4)**
>
> **W6: Whether the predicted evidence regions align with the model's internal attention patterns or decision process.**
>
> **A6**: We thank the reviewer for this insightful suggestion, as validating the link between predicted evidence and the model's internal process is key to our interpretability claims. We conducted precisely this validation, with qualitative results presented in **Figure 15 in our revised version.**
>
> To investigate this, we visualized the model's internal attention maps. The attention values are the mean attention of generated tokens related to bounding boxes (highlighted in red). Strong correlations are observed across examples, indicating that the model's focus is truly guided by those textual bounding boxes-based hints. This explains why visual grounded reasoning pipelines could be so effective. Note that sometimes the visual attention is still being overwhelmed by textual domain knowledge even if the grounding is accurate (see the failure case in Figure 15). Explicitly enforcing the model's focus beyond attention may be one potential solution.
>
> Related conclusions are also reported in [7] (Figure 5 in its paper), where the authors find that visual grounded reasoning maintains persistent visual attention throughout the reasoning process by dynamically localizing and incorporating relevant visual regions.
>
> **References**
>
> [1] LIME: Less Is More for MLLM Evaluation. ACL, 2025.
>
> [2] TinyBenchmarks: evaluating LLMs with fewer examples. ICML, 2024.
>
> [3] 100 instances are all you need: predicting the success of a new LLM on unseen data by testing on a few instances. KDD, 2024.
>
> [4] Not All Data Are Unlearned Equally. COLM, 2025.
>
> [5] Glm-4.1 v-thinking: Towards versatile multimodal reasoning with scalable reinforcement learning. arXiv, 2025.
>
> [6] DeepEyesV2: Toward Agentic Multimodal Model. arXiv, 2025.
>
> [7] VLM-R3: Region Recognition, Reasoning, and Refinement for Enhanced Multimodal Chain-of-Thought. arXiv, 2025.

---

### Official Review · Reviewer_rrGV · 2025-10-31

**Soundness:** 3
**Presentation:** 3
**Contribution:** 3
**Rating:** 6
**Confidence:** 4

**Summary:**

This paper presents TreeBench, a benchmark for evaluating traceable visual grounded reasoning, and TreeVGR, a reinforcement learning framework that jointly supervises reasoning and localization using a dual IoU reward. TreeBench includes 405 expert-curated VQA samples with bounding box annotations to assess perception, reasoning, and evidence traceability. TreeVGR improves visual reasoning through a two-stage training process with supervised initialization followed by reinforcement learning, achieving notable improvements across multiple benchmarks and contributing to more explainable multimodal reasoning.

**Strengths:**

- Studying visual grounding in the reasoning process is very meaningful because most models may ignore intermediate results in decision-making and fail to learn the true causal relationships.
- A new benchmark has been constructed, which allows researchers to consider a wider range of factors.

**Weaknesses:**

- Although the paper emphasizes achieving traceability through bounding boxes, the "intermediate interpretability" of the inference chain is still weak if it relies solely on box localization measure (mIoU).
- Although the paper includes extensive comparisons with open-source multimodal models such as LLaVA-OneVision and Qwen2.5-VL, these models are not specifically designed for reasoning or reinforcement learning–based visual grounded reasoning. As a result, the experimental comparison may not fully demonstrate TreeVGR’s advantage over other reasoning-oriented approaches.
- In the *Reinforcement Learning with Traceable Evidence* stage, the paper assumes that reasoning chains should explicitly include bounding boxes to ensure traceability. However, the necessity of this design is not fully justified. Reasoning transparency could also be achieved through implicit or attention-based grounding without inserting explicit box tokens. The paper lacks an analysis or ablation to clarify whether explicit box supervision is essential for reasoning quality.
- It is recommended to discuss work related to the interpretability of visual grounding [1].

[1] Interpreting Object-level Foundation Models via Visual Precision Search. CVPR 2025.

**Questions:**

Please see the weaknesses

---

> ### Author Response · Authors · 2025-11-19
> **Rebuttal to Reviewer rrGV (Part 1)**
>
> We would like to thank Reviewer rrGV for your valuable time and insightful comments. We are especially grateful for your recognition that studying visual grounding in the reasoning process is "very meaningful", and for the insightful remark on the importance of learning true causal relationships. We are also pleased that you acknowledged our contribution in constructing a new benchmark that allows researchers to consider a wider range of factors.
>
> **W1: The "intermediate interpretability" of the inference chain is still weak if it relies solely on the box localization measure (mIoU).**
>
> **A1**: We sincerely appreciate this insightful comment, as it aligns with our long-term goal of enhancing the step-by-step interpretability of visual grounded reasoning, a key concern we also recognize in advancing traceable multimodal reasoning. We fully agree that relying solely on bounding box mIoU does not capture the full complexity of "intermediate interpretability" for inference chains.
>
> However, we emphasize that **mIoU serves as a critical, concrete anchor for linking visual evidence to intermediate reasoning steps** in the context of our work. It is at least one effective way to quantify the quality of reasoning traces.
> - For TreeBench's design (centered on "traceable evidence via bounding boxes"), mIoU directly reflects **whether the model's intermediate localization decisions align with *human-annotated* visual cues** (e.g., in "Perspective Transform" tasks, the model must first localize the observer and target object before inferring direction, and thus mIoU quantifies whether these two fundamental localization steps are accurate or not, which directly explains errors in the final reasoning result).
> - In TreeVGR's RL pipeline, the dual IoU reward explicitly enforces that the model's **intermediate reasoning trajectory is grounded in precise visual regions** (e.g., when answering "Which object is occluding the cat?", the model must first localize the cat and the occluding object, and mIoU ensures these two intermediate steps are not "black boxes", allowing us to diagnose whether errors stem from mislocalization or flawed logical inference, as shown in our Appendix H's failure case analysis).
>
> We **do *not* claim mIoU is the "most interpretable" measure** for intermediate reasoning. Instead, it is a **pragmatic and well-aligned choice** for our work: it directly leverages TreeBench's unique bounding box annotations (a core advantage over existing benchmarks) and provides a quantifiable link between visual grounding and intermediate reasoning.
>
> **W2: About TreeVGR's advantage over other reasoning-oriented approaches.**
>
> **A2**: We have compared our TreeVGR with two representative open-source reasoning-oriented visual grounded reasoning models, i.e., DeepEyes and Pixel-Reasoner, on **Tables 2, 3, and 4**, where TreeVGR consistently outperforms them significantly.

---

> ### Author Response · Authors · 2025-11-19
> **Rebuttal to Reviewer rrGV (Part 2)**
>
> **W3: About the effectiveness of explicitly outputting bounding boxes during reasoning.**
>
> **A3**: We agree that addressing the "necessity of explicit bounding box supervision" is critical to validating our design, especially given the valuable potential of implicit attention-based grounding. We would like to clarify two core points: (1) why implicit methods are less aligned with our "traceable reasoning" goal, and (2) how explicit box supervision provides unique, irreplaceable value (beyond performance gains), supported by our existing ablation and analysis:
>
> **(1) Implicit attention-based grounding is less suitable for "traceable evidence evaluation".** We fully agree that implicit attention mechanisms (e.g., attention maps in VLMs) can enable basic grounding. However, they fall short of our work's key requirements: **quantifiable, human-verifiable traceability of intermediate reasoning steps**, a cornerstone of TreeBench and TreeVGR. Implicit attention lacks two critical properties for this goal:
> - **Explicitness & verifiability**: Attention maps are often sparse, noisy, and hard to map to concrete visual regions (e.g., an attention peak does not clearly correspond to a "target object" or "intermediate reasoning step"). In contrast, **explicit bounding boxes provide a precise, human-annotated reference** to judge if the model's reasoning is grounded in the right visual evidence (e.g., in "Contact and Occlusion" tasks, the model must explicitly localize both the occluding and occluded objects. Attention maps cannot confirm if the model actually focused on these two specific regions, but bounding boxes can).
> - **Error diagnosability**: A core motivation of our work is to **"diagnose why a model fails" (e.g., mislocalization v.s. flawed logic)**. Implicit attention cannot distinguish these two errors (e.g., low reasoning accuracy could stem from unfocused attention or bad logic). Explicit boxes, however, let us link errors to localization quality (via mIoU). Failure cases demonstrated in Appendix H show both localization error and logical error. This diagnostic capability is impossible with implicit grounding and directly supports our "traceable reasoning" goal.
>
> **(2) Our ablation (Table 6 in the original manuscript) already validates the "necessity" of explicit box supervision for reasoning quality.** Row 3 (TreeVGR: with explicit box supervision) vs. Row 4 (w/o Traceable Evidence: no explicit box supervision, i.e., implicit grounding-like setting): Explicit box supervision brings +12.4 accuracy on TreeBench, +3.2 on V* Bench, and +3.3 on MME-RealWorld-Lite. The drastically larger gain on TreeBench (+12.4) is critical. It shows that when reasoning requires fine-grained, traceable visual evidence (TreeBench's core strength), implicit grounding (Row 4) fails to capture the necessary visual cues. In contrast, explicit boxes force the model to anchor reasoning to concrete regions, which is essential for handling TreeBench's hard cases (e.g., small targets, perspective shifts).
>
> **(3) Explicit box supervision aligns with real-world "thinking with images" needs.** Currently, explicit box output has become a standard for state-of-the-art VLMs (e.g., Qwen3-VL, OpenAI-o3) not just for performance, but because it enables human-in-the-loop verification, a key requirement for real-world multimodal reasoning (e.g., medical imaging, autonomous driving). Implicit attention cannot support this, as humans cannot easily validate if a model’s "implicit focus" is correct.
>
> In summary, explicit box supervision is not just "effective". It is **necessary** for our work's core goal of "traceable visual grounded reasoning", as it solves implicit methods' limitations in verifiability, error diagnosability, and alignment with real-world needs.
>
> **W4: Comparing with [1].**
>
> **A4**: While both TreeVGR and [1] engage with multimodal vision tasks, they diverge fundamentally in their core goals, frameworks, and evaluations. [1] is centered on interpreting pre-trained object-level foundation models (e.g., Grounding DINO) by generating precise attribution maps to explain model decisions and failures. In contrast, TreeVGR focuses on enhancing the visual grounded reasoning capabilities of LMMs through a traceable training pipeline, addressing gaps in complex, evidence-driven reasoning that extend beyond simple object localization.
>
> [1] specializes in object-level tasks, including object detection, visual grounding, and zero-shot detection for rare classes. Its design is tailored to explain model outputs for these specific tasks, with analysis limited to whether models correctly localize or classify objects.
>
> TreeVGR focuses on vision-centric reasoning that goes beyond simple localization. It addresses tasks such as perspective transformation, object contact/occlusion, spatial containment, and cross-object comparison. These tasks demand integrating visual evidence with logical inference, rather than just identifying object regions.

---

### Official Review · Reviewer_PkSa · 2025-11-01

**Soundness:** 3
**Presentation:** 3
**Contribution:** 2
**Rating:** 6
**Confidence:** 3

**Summary:**

This paper aims to enhance the "thinking with images" capability of large multimodal models (LMMs). To this end, we introduce a benchmark named TreeBench and a training pipeline called TreeVGR. Specifically, TreeBench comprises 406 visual question-answer pairs, each accompanied by trace evidence that serves as a verifiable grounding instance. The proposed TreeVGR method extends the reinforcement learning algorithm GRPO to incorporate relevant visual instances (e.g., bounding boxes) as a form of Chain-of-Thought. Extensive experiments on TreeBench and the V* benchmark demonstrate the effectiveness of TreeVGR.

**Strengths:**

1. The TreeBench Benchmark: They construct a novel VQA benchmark wherein each question is explicitly linked to a groundable instance that serves as traceable evidence, ensuring verifiability.

2. The TreeVGR Method: They propose a GRPO-based training pipeline designed to enhance the groundedness of LMMs in VQA. The TreeVGR method guides the model to identify and utilize relevant evidential instances as a Chain-of-Thought.

**Weaknesses:**

1. The paper does not explicitly measure the quality of the groundable evidence in TreeBench. Given the semi-automatic annotation process (Lines 91-101), how is the correctness of this evidence guaranteed? A compelling way to validate the importance of the evidence would be to observe if masking the critical instances (e.g., the bounding boxes) leads to a significant drop in VQA performance.

2. The overall reward $R=R_{acc} + R_{iou} + R_{format}$ combines terms from different scales. Were these reward components normalized to a common range to prevent any single term from disproportionately dominating the optimization?

**Questions:**

1. How is the causal relationship between the evidence and the final answer validated? A critical test would be to see if the model's performance drops significantly when the key evidence instances are masked.

2. Should the individual reward components ($R_{acc}, R_{iou}, R_{format}$) be normalized to the same scale? If not, there is a risk that the term with the largest magnitude could dominate the entire training process.

**Details Of Ethics Concerns:**

the paper doesn't need ethic review

---

> ### Author Response · Authors · 2025-11-19
>
> We sincerely thank Reviewer PkSa for the positive and constructive feedback. We are particularly encouraged that you recognize the value of our TreeBench benchmark, highlighting its novelty and its design that ensures verifiability through traceable evidence. We are also grateful for your acknowledgment of our TreeVGR method, noting its ability to enhance the groundedness of LMMs by guiding the model to use evidential instances as a Chain-of-Thought. Next, we provide point-to-point responses to each of your comments.
>
> **W1: About the quality of groundable evidence in TreeBench.**
>
> **A1**: (1) We would like to clarify that **all groundable evidence, i.e., bounding boxes of target instances, of TreeBench are annotated by human experts**. The semi-automatic process is only utilized when generating candidate questions, choices, and answers. After selecting or manually conducting an appropriate candidate triplet, *the human expert will define the target instance and annotate the bounding box for each instance.* Moreover, we further incorporate a round of quality control (L808-809) to ensure the correctness of the answer and the bounding boxes. (2) Following your valuable suggestion, we mask all instances during inference on TreeBench.  *The results below show a significant performance drop across all models.* This confirms that the bounding boxes in TreeBench are not only high-quality but also indispensable for accurate visual grounded reasoning, directly validating the importance of the evidence as you suggested.
>
> We have added the experiment to the revised manuscript (Table 5 in Appendix D) to make these points more explicit.
>
> | Mask Target Objects | Qwen2.5-VL-7B | InternVL3-8B | GPT-4o | o3   | Gemini-2.5-Flash | Gemini-2.5-Pro |
> |-|-|-|-|-|-|-|
> | No | 37.0 | 38.8 | 46.9 | 54.8 | 45.9 | 54.1 |
> | Yes | 31.8 | 29.6 | 29.1 | 33.8 | 29.9 | 33.1 |
>
> **W2: About the scale of each reward.**
>
> **A2**: Actually, each reward ranges from 0 to 1, either binary ($R_{format}$ and $R_{acc}$) or continuous ($R_{IoU}$), eliminating the need for additional normalization to unify scales, as elaborated below:
> - $R_{acc}$ (accuracy reward) and $R_{format}$ (formatting reward) are binary indicators: they take a value of 1 if the final answer is correct (or the output format meets requirements, i.e., enclosed in specified tags) and 0 otherwise, inherently fitting the [0,1] range.
> - $R_{IoU}$ (dual IoU reward) is a continuous metric calculated as the average of recall-IoU $R_{IoU}^R$ and precision-IoU $R_{IoU}^P$ (Equations 2-4). Both $R_{IoU}^R$ and $R_{IoU}^P$ measure the overlap between predicted and ground-truth bounding boxes, with each ranging from 0 (no overlap) to 1 (perfect overlap). Their average $R_{IoU}$ thus also falls into [0,1], ensuring consistency with the other two reward components.
>
> This ensures no single component disproportionately dominates the optimization. Therefore, we simply add them without any normalization terms. By the way, the GRPO performs batch-level automatic reward normalization during training. This step mitigates potential minor fluctuations in reward distribution across batches, further guaranteeing balanced optimization of all three components.

---

### Author Response · Authors · 2025-11-19
**General Response and Paper Revision**

We would like to express our sincere gratitude to all reviewers for their insightful and constructive feedback on our work. We are particularly encouraged that:
- **Reviewers recognized the significance of our research problem.** **Reviewer rrGV** noted that studying visual grounding in the reasoning process is "**very meaningful**". **Reviewer qSgY** appreciated that our work "**identifies a gap in current multimodal research**" and frames the need for traceable visual reasoning in a straightforward way.
- **All reviewers unanimously acknowledged the contribution of our new benchmark, TreeBench.** It was collectively described as a **"novel" (PkSa)**, **"carefully constructed" (qSgY)**, and **"manually verified" (qSgY)** benchmark. We are glad that reviewers found it **"fills a gap" (o6mU)** in existing benchmarks by testing a model's ability to attend to fine-grained details and provides traceable evidence, ensuring verifiability.
- **Our proposed method, TreeVGR, was also well-received for its effectiveness.** Reviewers highlighted that it **"significantly improves the performance" (o6mU)** and **"enhances the groundedness" (PkSa)** of the model. We are also pleased that **Reviewer qSgY** found the pipeline to be **"simple, reproducible, and effectively demonstrates"** the value of our approach. Finally, we appreciate **Reviewer qSgY**'s commendation on the clarity of our writing and the consistent reporting of our results, which they believe makes our work **"easy to understand and potentially useful for future follow-up work"**.

We have tried our best to revise the paper to address all concerns. **All revisions are marked in purple.** Specifically:
- **@Reviewer PkSa**, we have added experiments with masked visual evidence in **Table 5**, where the results show a significant performance drop across all models. This confirms that the **bounding boxes in TreeBench are not only high-quality but also indispensable for accurate visual grounded reasoning.**
- **@Reviewer qSgY**, we have added experiments with explicit bounding boxes-based textual hints in **Table 6**, where all models achieve consistent performance gains, indicating that **bounding boxes help models explicitly anchor their reasoning to visual evidence.**
- **@Reviewer qSgY**, we have added qualitative examples between the output bounding boxes and the internal attention values in **Figure 15**, where strong correlations are observed across examples. This evidence demonstrates that **the model's attention is truly guided by those textual bounding box-based textual hints.**

Please let us know if you have any further questions. We are always looking forward to open discussions. We will give our response as soon as possible once you raise more questions.

Sincerely,

Authors

---

### Meta-Review · Area_Chair_VXvr · 2026-01-04

**Summary:**

This work addresses grounding of large multimodal models and there ability to deal with images. The paper introduces a new benchmark and an associated training pipeline. The work has received reviews of four experts and was on the fence with ratings of 6,6,4,4.

The reviewers agreed on the general usefulness of the benchmark contribution and the idea of the training pipeline. However, numerous weaknesses were raised:

- semi-automatic annotation process
- Scaling of the manual GT gathering process - the benchmark consists of 406 visual question-answer pairs only.
- choice of base models wrt to the claimed contribution
- lack of diversity in tasks
- possibility of overfit on reasoning tasks
- Missing ablations and missing evaluations
- Missing references to prior work
- Unclear origins of gains
- Results are provide on self-constructed benchmark only / results on public benchmarks are missing

There was visible discussion which survived the program chairs' actions after the openreview leak and the reassignment of this submission to the new AC, so there is no information how the ratings would have been modified given the numerous authors' answers. The AC would like to benefit from the opportunity to scold the authors on the sheer amount of information provided here - be brief! You are not doing yourself and certainly nobody else a favor be providing tons of text. There is an art in writing short answers.

The AC went over the authors' answers and found them convincing. Many of the weaknesses were misunderstandings or requests for experiments which were actually in the original paper (Just 2 examples: the semi-automatic process is actually manual; results on public datasets were given). Other requested results were provided and convinced the AC.

The only really remaining downside of this work seems to be the smallish side of the benchmark. 406 pairs is really not much, and in the ACs mind this will quite limit the impact of the benchmark part of the work.

In spite of this, the AC considers that this work is valuable for the community and recommends acceptance.

**Reviewer Concerns:**

Described in the meta review

**Reviewer Scores:**

Described in the meta review

---

### Decision · Program_Chairs · 2026-01-26

Accept (Poster)